# Similar temperature scale for valence changes in Kondo lattices with different Kondo temperatures

K. Kummer[1], C. Geibel [2], C. Krellner [2,3], G. Zwicknagl[4], C. Laubschat[5], N.B. Brookes[1] & D.V. Vyalikh[6,7,8]

The Kondo model predicts that both the valence at low temperatures and its temperature dependence scale with the characteristic energy $T_K$ of the Kondo interaction. Here, we study the evolution of the $4f$ occupancy with temperature in a series of Yb Kondo lattices using resonant X-ray emission spectroscopy. In agreement with simple theoretical models, we observe a scaling between the valence at low temperature and $T_K$ obtained from thermo-dynamic measurements. In contrast, the temperature scale $T_v$ at which the valence increases with temperature is almost the same in all investigated materials while the Kondo temperatures differ by almost four orders of magnitude. This observation is in remarkable contradiction to both naive expectation and precise theoretical predictions of the Kondo model, asking for further theoretical work in order to explain our findings. Our data exclude the presence of a quantum critical valence transition in $YbRh_2Si_2$.

[1] European Synchrotron Radiation Facility, 71 Avenue des Martyrs, CS 40220, 38043 Grenoble Cedex 9, France. [2] Max Planck Institute for Chemical Physics of Solids, Nöthnitzer Strasse 40, 01187 Dresden, Germany. [3] Kristall- und Materiallabor, Physikalisches Institut, Goethe-Universität Frankfurt, Max-von-Laue Stasse 1, 60438 Frankfurt am Main, Germany. [4] Institute for Mathematical Physics, TU Braunschweig, Mendelssohnstraße 3, 38106 Braunschweig, Germany. [5] Dresden University of Technology, Institute of Solid-State and Materials Physics, 01062 Dresden, Germany. [6] Saint Petersburg State University, Saint Petersburg 198504, Russia. [7] Donostia International Physics Center (DIPC), Departamento de Fisica de Materiales and CFM-MPC UPV/EHU, 20080 San Sebastian, Spain. [8] IKERBASQUE, Basque Foundation for Science, 48011 Bilbao, Spain. Correspondence and requests for materials should be addressed to K.K. (email: kurt.kummer@esrf.fr)

While most of the rare earth (RE) elements show a very stable trivalent state in solids, some of them, especially Ce, Yb, and Eu, may display different valence states depending on the chemical composition, the temperature, and the applied pressure or magnetic field[1]. This valence instability, which is caused by the $f$ shell being almost empty (Ce), almost full (Yb) or almost half-filled (Eu), is connected with a hybridization between localized $4f$ and delocalized valence states[2]. Weak hybridization results in a stable trivalent RE state with large local magnetic moments which order at low $T$ due to RKKY exchange. An increasing hybridization first results in an effective anti-ferromagnetic exchange interaction between the conduction electrons and the local moments, called Kondo interaction[3, 4]. As a result the local moments gets screened by the conduction electrons, the magnetic order disappears, leaving a paramagnetic ground state with strongly renormalized conduction electrons, referred to as Kondo lattices and heavy fermion systems. In these systems the occupation of the $f$ shell, i.e. the number of $4f$ electrons $n_f$ for Ce or $4f$ holes $n_h$ for Yb, shows a weak decrease with increasing hybridization and the valence $v$ starts to deviate from the trivalent state $n_{f,h} = 1$. Models based on the Kondo interaction have proven to be very appropriate to describe this regime[5, 6]. In the limit of strong hybridization, one enters the intermediate valence regime where a strong quantum mechanical mixing of different valence state occurs and real charge fluctuations are dominant[1, 2]. Theoretically this regime can be described with the Anderson model, which takes both the large Coulomb repulsion between $f$ electrons and the hybridization between $4f$ and conduction electrons explicitly into account and allows for both spin and charge fluctuations. In this case, $n_f$ and the valence can considerably deviate from 1+ and 3+, respectively. Since the competition between the RKKY interaction on one hand and the Kondo interaction and valence fluctuations on the other hand leads to very unusual states and properties, such systems have attracted a lot of attention[7–10]. Particularly fascinating are phase transitions at absolute zero temperature, called quantum phase transitions, which are often connected to unconventional superconductivity. Knowledge of the RE valence and of the occupation of the $f$ level is crucial for an understanding of the exotic properties of such systems.

The simplicity of the X-ray absorption/emission measurements at the $L_{2,3}$ edges of RE elements and the large X-ray penetration depth providing bulk specific results made this technique a standard for the study of the valence in Kondo lattice and intermediate valent systems as a function of composition and temperature. Hence a large number of studies have been performed to determine $n_f$ as well as the Kondo or valence fluctuating energy scale using the $T$ dependence of $n_f$[11–19]. However, the accuracy to which $n_f$ could be determined with standard XAS or photoemission measurements was limited to maybe a few percent[11]. Since in Kondo lattices the deviation from the trivalent state is small and typical changes in $n_f$ between 3 and 300 K are only of the order of a few percent, these studies of the $T$-dependent changes concentrated on systems with a relatively high $T_K \geq 100$ K. However, the most interesting cases are those where the deviation from trivalent is small, since this is the regime where the transition from a magnetic ordered to a paramagnetic ground state occurs[3, 20]. The strong development of resonant X-ray emission spectroscopy (RXES) has opened new possibilities for studying even minute valence changes in Kondo lattices and to determine $n_f$ with very high accuracy[11, 21]. The Yb case is especially suited for RXES because the weak signal of the "minority" Yb$^{2+}$ state is on the low energy side of the absorption edge, i.e. in a region with very low background by other emission lines. RXES allows to selectively enhance this Yb$^{2+}$ signal by at least one order of magnitude[21].

The extended and detailed knowledge accumulated on the Kondo alloy systems Yb(Rh$_{1-x}$Co$_x$)$_2$Si$_2$ and Yb(Rh$_{1-x}$Ir$_x$)$_2$Si$_2$ makes them perfect candidates for a systematic investigation of the relation between $n_f$ and the Kondo scale[20, 22–25]. With increasing Co doping the Kondo scale is gradually reduced by almost two orders of magnitude from about 25 K for YbRh$_2$Si$_2$ to less than 1 K for YbCo$_2$Si$_2$[22]. In contrast substituting Ir for Rh increases $T_K$ by about a factor of two up to 45 K[26, 27]. YbNi$_4$P$_2$ is a Kondo lattice system just at the verge of the transition from a ferromagnetic to a paramagnetic ground state, with a Kondo temperature of 8 K which increases upon substituting As for P[28, 29].

We have used the high accuracy offered by this method for a high precision study of the $T$ dependence of $n_f$ and its relation to bulk properties in Yb-based Kondo lattices. Our experimental data reveal an almost perfect scaling of the low temperature valence with $T_K$, in agreement with theoretical predictions. In contrast, the temperature scale on which the valence changes occur is almost the same in all compounds despite the fact that the Kondo temperatures differ by several orders of magnitude. This unexpected finding is in stark contradiction to predictions of the Kondo model. Comparing with the data on other Yb Kondo lattices that are available in the literature, we find that this behavior is observed in all of them and even seems to extend into the mixed-valent regime. We discuss the influence of excited crystal field levels and lattice vibrations as possible origins of this unexpected observation, but substantial theoretical work seems necessary in order to explain our findings.

## Results

**Resonant X-ray emission spectroscopy.** In Fig. 1 the obtained RXES spectra as a function of temperature are shown for four

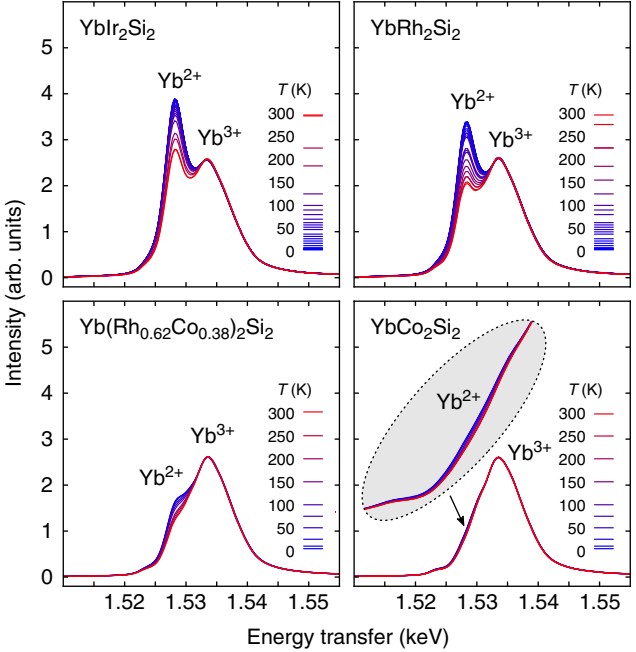

**Fig. 1** Yb $L_3$-$L\alpha_1$ RXES spectra as a function of temperature. Data are shown for four Yb Kondo lattice systems with different $T_K$ from 45 K (YbIr$_2$Si$_2$) down to well below 1 K (YbCo$_2$Si$_2$). The incident photon energy was set to the maximum of the 2+ resonance at the Yb $L_3$ absorption edge. All spectra are normalized to the intensity of the Yb$^{3+}$ emission line. The inset shows a zoom-in on the Yb$^{2+}$ region of the YbCo$_2$Si$_2$ spectra. Even though $T_K$ is well below 1 K in this compound, small but systematic changes of the Yb$^{2+}$ intensity all the way up to room temperature are observed

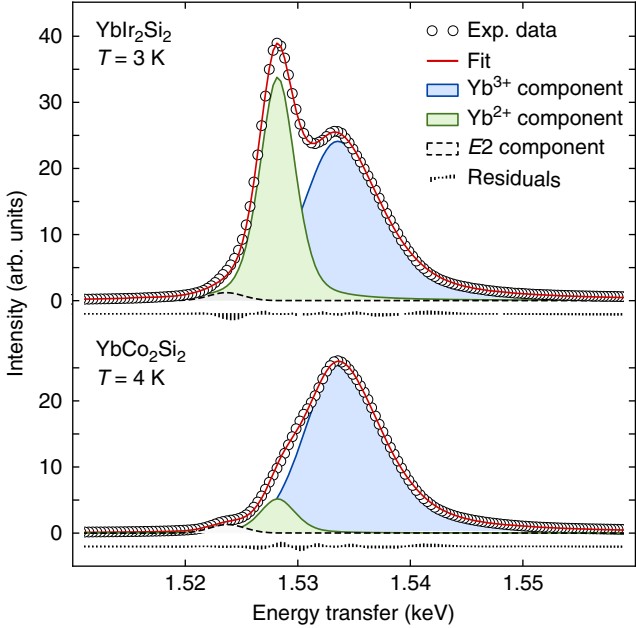

**Fig. 2** Quantitative analysis of the RXES spectra. The spectra are fitted using three components corresponding to the $Yb^{3+}$ (blue filling) and $Yb^{2+}$ (green) emission lines and the weak quadrupole transitions $E2$ (white) in the pre-edge region, respectively[21]. The line shape of the $Yb^{3+}$ and $Yb^{2+}$ lines is not symmetric and has been determined from the experimental data by looking at the difference spectrum between high and low $T$ data (see Supplementary Note 1). The Yb valence is then determined from the ratio of the $Yb^{2+}$ and $Yb^{3+}$ integrated intensities knowing that the $Yb^{3+}$ emission line is reduced to 13% of its maximum intensity when the incident photon energy is set to the $Yb^{2+}$ resonance of the $L_3$ absorption edge (see ref. [21] and Supplementary Note 2). The same lineshapes are used to fit the $Yb^{2+}$ and $Yb^{3+}$ components of the RXES spectra for all samples and at all temperatures

selected compounds. The data have been normalized to the peak intensity of the $Yb^{3+}$ emission line. For $YbIr_2Si_2$ and $YbRh_2Si_2$ with a relatively high Kondo temperature the temperature effects are sizeable and easily seen by eye. In $YbCo_2Si_2$ where the Kondo scale is small the deviation from integer valent $Yb^{3+}$ is very small and consequently the intensity of the $Yb^{2+}$ line is very weak. However, even in this case the variation of the $Yb^{2+}$ intensity with temperature can be measured and quantified with extremely high accuracy thanks to the excellent performance of the spectrometer yielding very high quality data. This is shown in Fig. 2 for the low temperature spectra of $YbIr_2Si_2$ and $YbCo_2Si_2$ and explained in detail in Supplementary Notes 1 and 2.

**Temperature dependence of the 4f occupancy**. The experimentally determined deviation from the trivalent (one hole) Yb state $\Delta n_h(T) = 1 - n_h(T)$ as a function of temperature is plotted in Fig. 3a for all studied Yb Kondo lattices. The absolute changes in the 4f occupancy and, in particular, the low temperature values $\Delta n_h(T \to 0) =: \Delta n_h(0)$ are consistent with the expectation that $\Delta n_h(0)$ increases as $T_K$ increases. Surprisingly however, the temperature scale $T_v$ on which the valence changes occur seems to be very similar in all measured compounds despite the very different Kondo scales, $T_K$. Here we identify $T_v$ with the temperature at which $\Delta n_h(T) = 1/2\Delta n_h(0)$ following ref. [30]. The dashed lines are fits to the data using the empirical form $\Delta n_h(T) = \Delta n_h(0)/[(1 + (2^{1/s} - 1))(T/T_v)^2]^s$, with $\Delta n_h(0)$ and $T_v$ free parameters and $s = 0.21$[31], which approximates the observed $T$

dependencies well. Note that other ways of assigning a scale $T_v$ to each curve in Fig. 3 can be used. This will change the absolute numbers for $T_v$ but it will not affect our finding that $T_v$ is very similar among all studied compounds (see Supplementary Note 6).

In order to be able to better compare the temperature dependence for the different samples we show the changes in $\Delta n_h(T)$ normalized to their low-temperature values in Fig. 3b. Apparently, there is no significant difference in the characteristic temperature scale $T_v$ of the valence changes neither within the Co doping series $Yb(Rh_{1-x}Co_x)_2Si_2$ nor with respect to the other measured Yb Kondo lattices. It should be noted that the temperature dependencies obtained here also resemble those reported in the literature for other Yb Kondo lattices, some with a $T_K$ as high as several hundred Kelvin[13–16, 19]. Also in these cases a clear $T$ dependence in the Yb valence between 3 and 300 K, the range most easily accessible to X-ray spectroscopies, was observed. For comparison we show those data in Fig. 3b together with our results. The plot clearly demonstrates that even though the Kondo scale differs by up to four orders of magnitude between the different compounds, the Yb valence shows a very similar $T$ response regardless of $T_K$. In contrast, the measured low temperature values of the Yb valence correlate very well with the Kondo scale determined in other experiments.

It should be noted that Yb-systems which are in the strong intermediate valent regime, i.e. with valence well below 3+, do not show a significant change in the valence in the $T$ range below 300 K. As an example we show in Supplementary Fig. 4 results for $YbAl_2$. This system presents a valence of about 2.2 and a characteristic valence fluctuation energy of the order of 2500 K[32]. The experimental RXES spectra at 3 and 300 K lie on top of each other. Thus any change in the valence is smaller than the experimental resolution. A further intermediate valent system for which precise RXES data exist, $YbAlB_4$, with a valence of the order of 2.75, shows a $T$ dependence of the valence below 300 K (Supplementary Fig. 4)[18], but the change is much weaker than in the systems shown above. This suggest that within the strong intermediate valent regime both the magnitude of the $T$-induced change in the valence, as well as the $T$ scale on which it occurs do change with the strength of charge fluctuations. This temperature behavior is much more in line with theoretical predictions according to which the characteristic temperature scale for valence changes in the mixed valent regime should increase only linearly with the bare hybridization strength $\Gamma$, and not exponentially as in the Kondo regime (c.f. Supplementary Note 3). However, our data on the $YbT_2Si_2$ series demonstrate that once the system enters from the strongly mixed-valent into the Kondo regime, $T_v$ stops to decrease and stays within a factor of two at a universal value, even when $T_K$ drops by four orders of magnitude. We note that this effect is also clearly visible on recent X-ray absorption spectroscopy (XAS) data on $YbNi_3Ga_9$ under pressure[33]. When the pressure is increased from $p = 0$ to $p = 16.1$ GPa, the valence at low $T < 10$ K increases continuously from $v = 2.6$ to $v = 2.88$, but the $T$ scale on which $v$ is increasing towards 3+ does not change significantly (Fig. 2 in ref.[33]).

Our findings are summarized in Fig. 4 where we plot for each Kondo lattice compound the experimentally measured low temperature deviation from the trivalent Yb state, i.e. $\Delta n_h(0)$, against $T_v$, the temperature scale of the valence changes, and $T_K$, the Kondo energy scale inferred from magnetic susceptibility, entropy or Mössbauer measurements[28, 34, 35]. For comparison we again included the results reported in the literature[13–16, 19, 35] in panel (b) and also (a) if $T_v$ has been determined from the X-ray spectroscopy data. While there is a clear correlation between $\Delta n_h(0)$ and $T_K$, $T_v$ seems to be independent of either of these two.

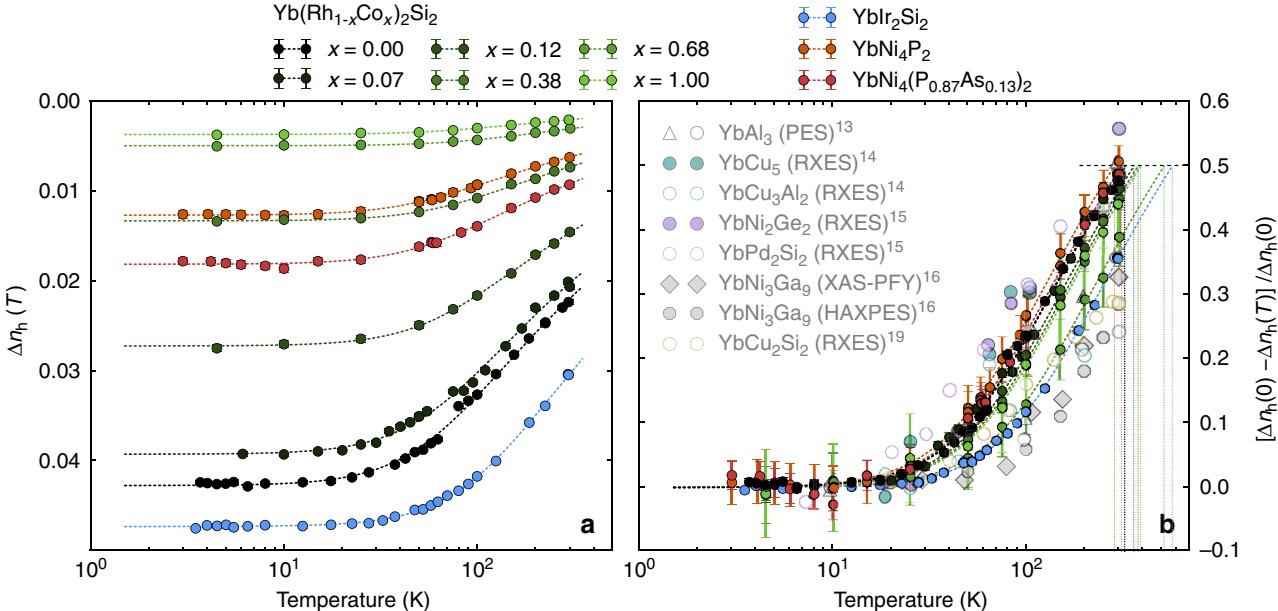

**Fig. 3** Evolution of the 4*f* occupancy with temperature. Temperature dependence of $\Delta n_h(T)$, the deviation of the Yb valence from 3+, in absolute values (**a**) and normalized to its low-temperature value (**b**). The error bars show the $2\sigma$ confidence interval calculated from the covariance matrix of the fits of the 2+ and 3+ amplitudes in the experimental spectra. The absolute error for $\Delta n_h(T)$ in **a** is comparable for all compounds and within the symbol size. The relative error in **b** increases with smaller values of $\Delta n_h(T)$ but stays within ±5% even for the smallest values of $\Delta n_h(T)$. The dashed lines are fits to the data points with an analytic expression from which the temperature scale of the valence fluctuations, $T_v$, and the low temperature value, $\Delta n_h(0)$, are obtained (see text for details). Previously reported results on other Yb Kondo lattices[13–16, 19] are shown in faded colors in **b** for comparison, together with the experimental technique that was used to obtain the data

## Discussion

Within the usually used approximations the Kondo model predicts that both the low temperature valence, $3 - \Delta n_h(0)$, and the temperature scale, $T_v$, on which the valence changes occur will increase with increasing Kondo scale $T_K$ (see e.g. ref. [30], n.b. Supplementary Note 5). Figure 4b clearly shows that there is indeed a very nice correlation between $\Delta n_h(0)$ measured with RXES and the thermodynamically determined $T_K$. The linear relation between log $\Delta n_h(0)$ and log $T_K$ suggests a power law dependence between both properties, i.e. $\Delta n_h(0) \approx aT_K^n$. The dashed line in Fig. 4b corresponds to $n = 2/3$ and $a = 1/200$. Within the Kondo model both $\Delta n_h(0)$ and $T_K$ depend on several parameters. Therefore the exact relation between both properties is a complex one. However, in a simple approximation, $\Delta n_h(0)$ is proportional to $T_K/\Gamma$, where $\Gamma$ stands for the bare hybridization width of the ground state doublet[30, 36]. $T_K$ itself depends exponentially on $\Gamma$ which results in a weak logarithmic dependence of $\Gamma$ on $T_K$ (see Supplementary Note 3). In total that should lead to a sublinear dependence of $\Delta n_h(0)$ on $T_K$. Therefore, given the approximations made, the $n = 2/3$ power law we observed is in reasonable agreement with theoretical expectations.

The observed perfect scaling between $\Delta n_h(0)$ and $T_K$ strongly indicates that the Kondo effect is the dominant interaction responsible for $\Delta n_h$. It is therefore a big surprise that the $T$-induced increase of the Yb valence towards 3+ occurs on almost the same temperature scale $T_v$ for all compounds (Fig. 4a), independently of $\Delta n_h(0)$ and $T_K$. The Kondo model has proven to be very successful in explaining even fine details of the properties of Yb-Kondo lattices[5, 6], but it predicts the $T$ scale of the recovery of the 3+ state to be proportional to $T_K$, an intuitively trivial relation (n.b. Supplementary Note 5). One therefore needs to look for possible mechanisms that could modify the $T$ dependence expected within the simple approximation usually used for Kondo systems.

A first candidate is the crystal electric field (CEF), which is not taken into account in those simple approximations, but which is known to be important in RE compounds. CEF leads to a splitting of the $J = 7/2$ multiplet of Yb$^{3+}$ into four doublets (or one quartet and two doublets in case of a cubic local Yb symmetry). Usually for Kondo physics only the lowest doublet is considered but at higher temperatures, when $T$ becomes comparable to the CEF splitting, excited CEF levels get thermally populated and become relevant. However, even for a single Kondo ion, an exact calculation of $n_h$ including CEF is extremely demanding, and therefore to the best of our knowledge has only been done for one very specific case[36, 37]. Hence the effect of CEF on the $T$ dependence of $n_h$ is not very clear at the moment. Naively, because the Kondo scale depends exponentially on the degeneracy $N$ of the effective local moment, involvement of excited CEF states corresponds to an increase in the effective degeneracy which should result in an increase of the effective Kondo scale with increasing $T$. Because of the exponential dependence of $T_K$ on $N$, this effect would become stronger upon decreasing $T_K$ of the ground state. Thus it would provide a way for making $T_v$ independent of the ground state $T_K$ (for a more detailed argument see Supplementary Note 4). A similar effect would also be expected if the hybridization of an excited CEF level is stronger than the hybridization strength of the ground state doublet. Different hybridization strengths for different CEF levels have rarely been considered so far. But recent theoretical and experimental results indicate that the hybridization depends on the symmetry of the CEF level[38, 39], and that this might result in unexpected phenomena[40, 41]. In YbRh$_2$Si$_2$, where the hybridization of the CEF ground state seems to be relatively weak, such an effect might become important. An STM study of YbRh$_2$Si$_2$ indeed provided evidence that in this compound the evolution of the Kondo lattice at higher temperatures is strongly influenced by excited CEF levels[42]. Different compounds might have very different CEF schemes, but the overall CEF splitting, i.e.

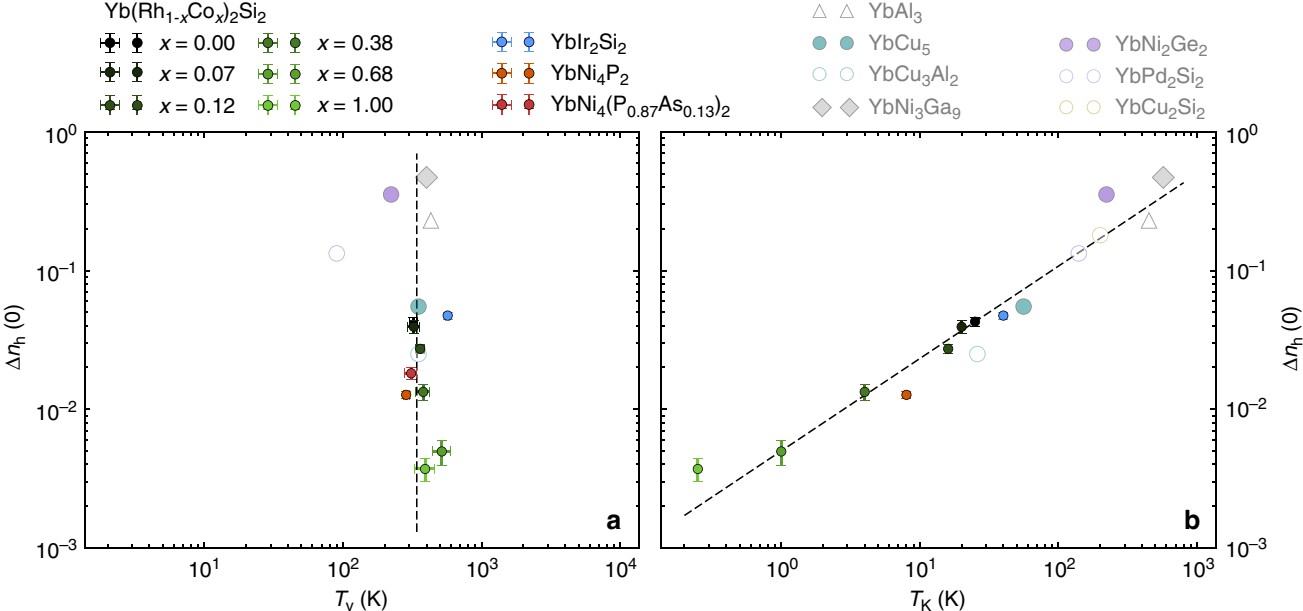

**Fig. 4** Dependence of the low temperature valence on characteristic temperature scales. The deviation of the Yb valence from 3+ at low temperatures, $\Delta n_h(0)$, is plotted against **a** the characteristic temperature scale of the valence changes, $T_v$, extracted from the data shown in Fig. 3, and **b** the Kondo energy scale, $T_K$, obtained from thermodynamic measurements. Data points taken from previous reports in the literature[13-16, 19] are shown in shaded colors. The dashed lines are guides to the eye. The data show that $\Delta n_h(0)$ scales with $T_K$ as expected, over several orders of magnitude, while $T_v$ does not scale with $\Delta n_h(0)$ or $T_K$ and is very similar in all investigated Yb Kondo lattices

the energy difference between the CEF ground state and the highest excited CEF level do not differ much and are typically in the range 200 to 400 K. Thus both the absolute value and the range of the overall CEF splitting are compatible with the value and range of $T_v$.

Another factor that could be relevant for $n_h(T)$ is the lattice. It seems that thermal expansion does not have a significant effect on $n_h(T)$. In Yb Kondo lattices increasing the volume results in a shift of the valence towards the 2+ state, because of its larger atomic volume[25]. Therefore normal thermal expansion should lead to an increase of $\Delta n_h(T)$ with increasing $T$, opposite to the experimental observation. On the other hand, $T_v$ is about 300 −400 K, which is similar to the Debye temperatures $\Theta_D$ of these compounds. $\Theta_D$ does not show large variations between the studied Yb compounds, typically differing by not more than ±30%. These differences would be even smaller if one would consider a $\Theta_D$ only corresponding to the vibration of Yb atoms. Thus it is tempting to discuss the universal $T_v$ scale in terms of a phonon-induced mechanism. The renormalization of the quasi-particles at the Fermi level $E_F$ due to electron−phonon coupling is indeed expected to vanish on the scale of $\Theta_D$ which would lead to a decrease of the renormalized electronic density of states at $E_F$ on the same $T$ scale. As the Kondo energy is proportional to the renormalized electronic density of states, one would therefore expect $T_K$ to decrease with increasing temperature on a $T$ scale related to the Debye temperature. But at the same time, the exchange coupling itself should also be affected by thermally excited phonons and a preliminary analysis indicates that it might increase with increasing $T$, thus compensating the effect of the change in renormalization. Therefore the effect of enhanced thermal vibration is not yet clear and requires more theoretical and experimental studies. On a general level, in strongly correlated electron systems, the effect of the lattice on the electronic properties is usually just ignored. However, recent studies indicate that taking the lattice into account might severely change the expected electronic properties[43]. Very recently, a study of the effect of zero point and thermal ionic motion on the

superexchange $J$ in cuprates indicates a strong motional broadening of $J$ giving rise to substantial effects on some electronic and magnetic properties[44].

Our results imply that a determination of the Kondo scale based on the $T$ dependence of the valence determined with X-ray spectroscopy, as frequently done, is at least questionable. Instead this $T$ dependence might not necessarily be related to the Kondo scale. RXES is intimately related to standard $L_{2,3}$ edge XAS and other X-ray and electron spectroscopic techniques. Previous studies comparing the different techniques showed that hard and soft X-ray photoemission yield the same low $T$ valences and $T$ dependences as X-ray absorption/emission techniques for Yb compounds[19, 21]. Part of the previously reported results on other Yb compounds which we included in Figs. 3 and 4 have been obtained with X-ray spectroscopies other than RXES and show the same $T$ dependence. Thus the surprising $T$-dependent behavior reported here is seen independent of the employed spectroscopic technique and seems to be universal among the investigated Yb Kondo lattice systems. It would be interesting to see whether the valence changes in Ce Kondo lattices show a similar temperature behavior independent of the Kondo scale $T_K$. Such a result might provide some further insight into the origin of the unexpected temperature dependence of $\Delta n_h(T)$.

The high sensitivity and accuracy in determining $\Delta n_h$ with RXES and the nice relation observed between $T_K$ and $\Delta n_h(0)$ suggest some immediate application in the field of Kondo lattices. In Kondo lattices where $T_K$ is of the order of or smaller than the magnetic ordering temperature, thermodynamic, transport and magnetic data cannot reliably determine $T_K$ since all these properties are then controlled by the RKKY energy scale. In such systems, a RXES measurement of $\Delta n_h(0)$ can provide a much more reliable estimation of $T_K$ using the power law scaling between $T_K$ and $\Delta n_h(0)$. This can be easily understood since $\Delta n_h(0)$, in contrast to many other properties, is not affected by the RKKY exchange.

The high accuracy of the present RXES data also allows settling a controversy in the field of Kondo lattice systems. YbRh$_2$Si$_2$ is

very close to the quantum critical point separating the anti-ferromagnetically ordered ground state from the paramagnetic ground state[45]. This results in very unusual thermodynamic, transport and magnetic properties, e.g. in a divergence of the effective mass of the conduction electrons[45]. Several models have been proposed to explain these unusual properties, but there is no consensus yet[46–48]. The topic is subject of an intense debate. One of the proposed models relies on the presence of a valence quantum critical point, i.e. a valence transition which is suppressed to $T = 0$ by pressure or chemical tuning[48]. However the present data and analysis demonstrate that throughout the alloy series $Yb(Rh_{1-x}Co_x)_2Si_2$ and including $YbIr_2Si_2$, the $T$ dependence of the valence is the same, and that the valence stays constant from lowest $T$ up to 20 K within a (relative) accuracy of $\Delta v \sim 5 \times 10^{-4}$. This excludes the presence of a valence transition, and thus the presence of a valence quantum critical point. The valence quantum critical point scenario of ref. [48] does not seem to be relevant for $YbRh_2Si_2$.

In summary, we used RXES to study the evolution of the valence in the Kondo alloy $Yb(Rh_{1-x}Co_x)_2Si_2$ and a few further Yb-based Kondo lattices as a function of composition and temperature. The Kondo temperature $T_K$ in the studied compounds cover the range from very small values $T_K \approx 0.25$ K up to quite large values $T_K = 40$ K. With present RXES techniques the Yb valence in such systems can be determined with very high accuracy. This allows us to study the deviation from a strict trivalent state, i.e. the deviation in the number of holes $n_h$ in the 4$f$ shell from unity, $\Delta n_h = 3 - v$, as a function of composition and temperature with a precision of the order of 0.1%. Combining the present results and literature data, we reveal an almost perfect linear correlation between $\log \Delta n_h(T = 0)$ determined from RXES at $T < 10$ K and $\log T_K$ determined from thermodynamic properties. This linear correlations extends over three orders of magnitude, from $T_K \approx 0.25$ K and $\Delta n_h(0) = 0.004$ up to $T_K \approx 500$ K and $\Delta n_h(0) = 0.4$. Upon increasing $T$, all systems evolve towards a pure trivalent state, but surprisingly the $T$ scale on which $\Delta n_h$ decreases with increasing $T$ is almost the same for all compounds, and thus does not scale at all with $T_K$. Looking for the origin of this unexpected universal $T$ scale we identify two possible candidates, excited CEF levels and lattice vibrations. However the effect of both properties on $n_h(T)$ has not been studied yet and is thus presently not clear. Therefore our results urge for more experimental and theoretical studies on how excited CEF levels and lattice vibrations affect the $T$ dependence of the valence. Without respective information available it is not clear whether excited CEF levels or lattice vibrations can explain our observations, or whether another mechanism is active. The very similar temperature scale $T_v$ observed in all compounds in this study gives, at least, a good idea of the energy scale of the relevant mechanism.

## Methods

**Sample details**. Single crystals of $Yb(Rh_{1-x}Co_x)_2Si_2$ ($x = 0.00, 0.07, 0.12, 0.38, 0.68, 1.00$), $YbIr_2Si_2$, and $YbNi_4P_2$ have been grown and characterized as described previously[49, 50]. The sharpness of the Laue diffraction pattern taken from the samples suggests high crystalline quality which for $YbRh_2Si_2$ and $YbCo_2Si_2$ is also confirmed by excellent ARPES and STM data[6, 42]. The physical properties of the samples have been extensively studied with e.g. specific heat, resistivity and magnetic susceptibility measurements. From these data a consistent picture of the Kondo scale in the $Yb(Rh_{1-x}Co_x)_2Si_2$ has emerged showing a gradual decrease of the single ion Kondo temperature $T_K$ from about 25 K for $x = 0$ to well below 1 K for $x = 1$[22, 25]. The onset of coherence effects in the resistivity which is typical for Kondo lattices scales similarly from about 120 K for $x = 0$ to ~ 1.5 K for $x = 1$.

**Resonant X-ray emission spectroscopy**. The RXES data have been taken with the X-ray emission spectrometer of the former ESRF beamline ID16, now upgraded and reopened as ID20, looking at the Yb $L_3$-$L\alpha_1$ emission line. Measuring with energy resolution in the scattered beam significantly reduces the life-time

broadening of the spectra and furthermore can be used to enhance the weak $Yb^{2+}$ contributions in the Kondo lattices[11]. For each sample we have taken the Yb $L_3$ XAS spectrum, then fixed the incident photon energy to the maximum of the $Yb^{2+}$ resonance and measured the Yb $L\alpha_1$ RXES spectrum. The experimental procedure and the data analysis are described in detail in ref. [21]. The data have been taken twice for each sample during two different experiments and using two different sample growth batches. The results of the first and the second run of experiments are exactly the same showing perfect reproducibility of our experimental findings. As a hard X-ray photon-in photon-out technique, Yb $L_3$ edge RXES is a bulk sensitive technique free of surface effects. The available temperature range was 3 −300 K and the data were taken starting from low temperature.

**Data availability**. The data that support the findings of this study are available from the corresponding author upon reasonable request.

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

## Acknowledgements

We thank L. Simonelli for assistance during the experiments at the ESRF. This work was supported by the German Research Foundation (DFG) (grants VY64/1-3, GE602/2-1, GE602/4-1, KR3831/5-1, Fermi-NEst, GRK1621 and SFB1143) and Saint-Petersburg State University under Research Grant 15.61.202.2015.

## Author contributions

K.K. and D.V.V. devised the research. C.K. and C.G. grew and characterized the samples. K.K. and D.V.V. performed the experiment. K.K. analyzed the data. K.K., C.G., and D.V. V. wrote the manuscript. The results and the manuscript were discussed with C.K., C.L., G.Z. and N.B.B.

## Additional information

**Competing interests:** The authors declare no competing interests.

