## [Peer Review File · Nature Communications]

Reviewers' comments:

Reviewer #1 (Remarks to the Author):

A long standing challenge in the field of strongly correlated electron materials that contain Yb (and other analogous elements such as Ce, Eu, U, and Pu) is the question of how f-electron valence instabilities influence low temperature phenomena. In particular, with respect to the appearance of phenomena such as breakdown of Fermi liquid behavior and unconventional superconductivity, their role near quantum phase transitions is important to sort out.

This manuscript aims to address these types of issues by focusing on the relationship between the Kondo energy scale T_K , the low temperature f-electron valence n , and the temperature dependence of the f-electron valence in several exemplary Yb-based correlated electron Yb-based materials including YbCo_2Si_2 , YbRh_2Si_2 , YbIr_2Si_2 , chemically substituted $\text{Yb}(\text{Rh},\text{Co})_2\text{Si}_2$, YbNi_4P_2 , and $\text{YbNi}_4(\text{P},\text{As})_2$. Comparisons are made to prior results for other Yb-based intermetallics with strong electronic correlations, thus allowing for exposition of trends amongst materials with Kondo temperatures ranging over several orders of magnitude. It is naively expected that n and its temperature dependence should scale with T_K . Here, resonant x-ray spectroscopy measurements suggest that while n scales with T_K , its temperature dependence does not. Possible explanations are proposed for this result, including that crystal electric fields splitting and/or lattice vibrations play important roles that were previously unappreciated.

The results for the low temperature saturating values of n and their relationship to T_K appear robust, and furthermore give a straightforward way to estimate T_K for systems where magnetic ordering occurs at or above the Kondo temperature. This is a valuable outcome of this study that will be of interest to researchers in this field.

On the other hand, the interpretation of the temperature dependence of n seems less reliable. Before considering non-obvious mechanisms such as crystal electric field splitting and lattice interactions, it is important to thoroughly convince the reader about the following points:

(1) The $\Delta n_h(T)$ data are non-saturating at high temperatures. How can we be sure about the fits to the data?

(2) A related issue that probably causes some uncertainty is that while the raw RXES data is sharp at low temperatures, it presumably broadens at high temperatures. How robust is the deconvolution of the data into the $2+$, $3+$, and background components at high temperature? Should there be error bars on n that grow with increasing T ? How does this change fits to $\Delta n_h(T)$?

(3) Just looking at Fig. 3b by eye, there is some variation in the temperature dependences of different materials. To what extent does the logarithmic x-axis submerge actual differences between the various materials?

If these points can be solidified, then there is still a question as to whether the proposed mechanisms for the anomalous temperature dependence are the right ones to consider. As stated by the authors, there is little work to help address this question. This makes it hard to assess the robustness of this discussion and makes it appear speculative.

To conclude, this manuscript attempts to approach an important problem in strongly correlated electron materials and finds a noteworthy (but not unexpected) relationship between n and T_K . This result even has a practical use with regards to characterizing materials where T_K is similar to or less than a magnetic ordering temperature. Unfortunately, the current presentation of the temperature dependent results and the subsequent interpretation are incomplete. With revisions, it might be possible to address the issues that are raised above and make this manuscript suitable

for publication in this journal.

Reviewer #2 (Remarks to the Author):

Please see the enclosed PDF file with the detailed comments to the Authors

Reviewer report, NCOMMS-16-29396-T

In the manuscript by Kummer *et al*, the authors have presented intriguing experimental evidence regarding the valence change as a function of temperature in several Yb mixed-valent compounds. Using highly sensitive resonant X-ray emission spectroscopy, they were able to measure the tiny deviations of the Yb valence from 3+. Overall, I find the experimental results very interesting and the apparent dichotomy between two panels of Fig. 4 startling. The manuscript is well written and I think would be of interest to the readership of Nature Communications. Having said this, I cannot communicate the manuscript for publication in its present state because in my opinion, the authors should first clarify several theoretical claims made in the manuscript. In particular, I challenge the authors' explanation of the near constancy of the T_v scale across different compounds (see below).

However first, I would like to start with a few questions about the experimental analysis used to extract the valence from the data in Figs. 1 and 2.

1. The authors claim at the bottom of page 4 that “the line shape of Yb³⁺ and Yb²⁺ lines is *not symmetric*”. How did the authors fit the data in Fig. 2? Did they use a skewed Gaussian and fitted the amount of skewing? Can they be sure that this procedure does not affect the numerically sensitive integral under the RXES curve in Fig. 1 and Fig. 2? In other words, what is the analytical profile of the curve used to fit Yb³⁺ and Yb²⁺ components in Fig. 2a and how accurately has it been determined?
2. Somewhat related: can the authors place the errorbars on the datapoints of the valence displayed in Fig. 3 and 4 as a result of numerical uncertainties in fitting the RXES spectra by (asymmetric) Gaussian-like curves? I was under the impression that providing the errorbars was a necessary prerequisite for publishing experimental data in the *Nature* family journals, is it not? I would imagine that the errorbars should be the larger, the smaller the value of $\Delta n_h(T)$ is, since the extraction of the valence relies on extracting a tiny shoulder in the Yb³⁺ curve, as is most pronounced in the case of YbCo₂Si₂ in Fig. 1.
3. On page 5, the authors make a vague statement that determining the ratio of Yb³⁺ and Yb²⁺ integrated intensities hinges, crucially, on the fact that, I quote “the Yb³⁺ emission line is reduced to *almost a tenth* of its maximum intensity when the incident photon energy is set to Yb²⁺ resonance of the L_3 absorption edge.” Can the authors please rewrite this sentence in a way that is legible by a non-expert? Do they mean that the integrated value under the Yb³⁺ line should really be multiplied by a factor of about ten? Is that what has been done to plot the data in Figs. 1 and 2? Why are the authors vague about the reduction factor? –“almost a tenth” sounds uncertain. I could argue that one ninth is also “almost a tenth,” so this would imply that there is a systematic 10% error in all the data? If the extraction of the valence hinges on this factor, it is important that the authors clarify it.

Now as for the theoretical analysis that is central to the authors' conclusions, I have a few questions:

4. Can the authors please formally define the quantity Γ first introduced at the bottom of page 8? I assume they meant $\Gamma = \pi|V|^2N(E_F)$, defined in terms of the hybridization V in the Anderson model, and $N(E_F)$ is the conduction electron DOS, is this correct? I would like to point out that in many other

literature sources, this quantity is denoted by Δ and is referred to as the “hybridization strength.” The authors should be wary of this and not assume that the reader will automatically understand the meaning of Γ .

5. I have a conceptual problem understanding the statement that “ T_K itself depends exponentially on Γ ” on page 8. You see, this is only true in the extreme limit of vanishing $U \rightarrow 0$ in the Anderson model, when Γ is at its bare value. Otherwise, the “hybridization width” becomes a running coupling constant, dependent on the energy scale, and some authors denote it by $\tilde{\Gamma}$ (or $\tilde{\Delta}$, in the notation of Alex Hewson), to distinguish from the bare value of Γ . In particular, in the “Kondo limit” $U \rightarrow \infty$, it is well known that the hybridization width is **not** given by the bare width $\Gamma = \pi|V|^2N(0)$ (this would be huge, of the order of ≈ 50 meV!); instead, $\tilde{\Gamma} \rightarrow k_B T_K$ (I refer the authors to Ch. 5 in Hewson’s book[1], where this is discussed in detail). So, the experimentally observed “hybridization width” is heavily renormalized and is actually *linearly* (not logarithmically!) dependent on the Kondo temperature T_K in this limit. I realize that the real system is not in the Kondo limit $U \rightarrow \infty$, but in some intermediate U regime; nevertheless, I would argue that the real Yb compounds are closer to the large- U limit than they are to the non-interacting limit. Therefore, it is not clear why the authors make a claim that the hybridization width is exponentially dependent on T_K . I urge the authors to re-evaluate their statement and to denote unambiguously which “hybridization width” they meant (and why).
6. Related to the above issue, the statement that “in a simple approximation, $\Delta n_h(0)$ is proportional to T_K/Γ ” is questionable. Can the authors please provide proof of this statement in either the Appendix or Supplementary Materials?
7. I take an issue with the authors’ statement on page 9 that the Kondo model “predicts the T scale of the recovery of the 3+ state to be proportional to T_K , an intuitively trivial relation.” I disagree, based on the following argument. Let me quote the exact result, based on the Friedel sum rule, that the occupancy of the f -level per spin channel is given by (see Eq. (5.47) in Ref. [1]):

$$\langle n_{f,\sigma} \rangle = \frac{1}{2} - \frac{1}{\pi} \arctan \left(\frac{\tilde{\varepsilon}_f}{\tilde{\Gamma}} \right) \quad (1)$$

where I’ve used the same notation as is standard in the Yamada–Yosida approach to the Anderson model (summarized in Ch. 5.3 of Hewson’s book [1]), modulo denoting the hybridization by Γ instead of Δ :

$$\tilde{\Gamma} = z\Gamma, \quad (2)$$

$$\tilde{\varepsilon}_f = z(\varepsilon_f - E_F + \Sigma_\sigma^R(E_F)), \quad (3)$$

$$z = (1 - \partial \Sigma_\sigma^R(E_F)/\partial \omega)^{-1} \quad (4)$$

It is difficult in general to obtain analytic expressions for the above quantities, however one approach that appears fruitful to me, is to use the poor-man’s RG scaling for the Anderson model, where some quasi-exact results can be obtained. In particular, I would like to quote the expression for $\tilde{\varepsilon}_f$ obtained specifically *in the mixed-valent regime*, from Eq. (3.73) in Hewson’s book:

$$\tilde{\varepsilon}_f = \varepsilon_f + \frac{\Gamma}{\pi} \ln \left(\frac{\pi D}{2 \max(\Gamma, k_B T)} \right). \quad (5)$$

The above equation captures the effect of the thermally excited charge fluctuations on the impurity site (i.e. mixed valence), which is what we are after. Substituting this into Eq. (1), one observes, qualitatively, that the change in the valence should occur at the temperature scale T_v when $k_B T_v \approx \Gamma$. Note that T_v is proportional to the bare hybridization scale $\Gamma = \pi|V|^2 N(E_F)$ of the Anderson model, and *not* the renormalized hybridization width $\tilde{\Gamma}$, and *not* the Kondo scale!

The key point is that the bare energy scale Γ is roughly constant across multiple Yb compounds and does not depend sensitively on the position of the f -level or the Kondo temperature. One concludes that the change in the valence from 3+ for Yb should occur at the temperature scale T_v that is roughly constant, of the order of $T_v \sim 500$ K (under the realistic estimate $\Gamma \sim 50$ meV). And this is exactly what the authors have found!

Of course I realize that the RG scaling argument is approximate and the authors may question the above derivation of Eq. (5). In its defense, I would like to show the data for n_f obtained by Pedro Schlottmann based on some exact results obtainable by Bethe ansatz, see Figure below. The different curves correspond to different bare positions ε_f of the f -level; in particular curves (c) and (d) are representative of the mixed valent regime. The key point is that regardless of exactly which curve one focuses on, the change in valence occurs at the temperature scale $T_v/\Gamma \sim 1$, proving my earlier point that T_v is of the order of the bare hybridization scale Γ and *not* the Kondo scale.

Figure 1: Dependence of the f -level occupation number on temperature in $N = 6$ Anderson model. Reproduced from Fig. 5.1 in Ref. [2]. Curves (c) $\varepsilon_f = 0$ and (d) $\varepsilon_f = -3\Gamma$ are representative of the mixed valent regime for Yb systems.

To summarize, while the experimental findings presented in the manuscript are certainly interesting, I challenge the authors' alleged interpretation of the valence scale T_v as being due to crystal-field effects, and I instead invite the authors to consider a much more mundane explanation, namely that the scale T_v is independent of the Kondo temperature for a simple reason that it is related to the bare hybridization scale $\Gamma \gg k_B T_K$, which is a lot less sensitive to the microscopic details of the composition, lattice constant etc.

References

- [1] A. C. Hewson, *The Kondo Problem to Heavy Fermions*, Cambridge Univ. Press (1993).
- [2] P. Schlottmann, *Phys. Reports* **181**, 1 (1989).

Reviewer #4 (Remarks to the Author):

content:

In this paper the authors measure the resonant X-ray emission spectra of YbRh₂Si₂ and related Yb compounds, and derive the relative weight of divalent and trivalent initial states as a function of temperature. By fitting the relative weight by a certain smooth function, the authors claim the presence of a common temperature scale in systems with very different Kondo temperatures. Possible origins of the common scale are discussed.

scientific validity:

The "universal temperature scale" is obtained by the procedure explained in Fig.3, where the data at temperatures between 3K and 300K are fitted by a certain smooth function. The authors deduce the temperature scale T_v around 300-400 K even though no data are available above 300 K. The most serious drawback of the paper is the procedure to derive T_v . Namely

(1) The authors assume trivalent Yb in the high temperature limit, which has no justification both experimentally and theoretically. Experimentally, only some increase of Yb valency with increasing temperature has been found.

(2) Extrapolation to temperatures higher than 300 K relies on a function with no physical basis. Hence the value of T_v obtained around 300-400 K is not trusted, and the major conclusion does not have a scientific basis. Although the authors cite the paper by Goldhaber et al (ref.31) that uses a similar function, the quantity fitted in ref.31 is the conductance with empirical validity for interpolating the temperature dependence.

Other remarks:

The analysis in this paper using the terminologies of "Kondo model" and the "Kondo interaction" is very confusing. In the standard usage, the Kondo model refers to the case where charge (valence) fluctuations are projected out. If the valence fluctuation is to be included explicitly, the Anderson model for example should be used that allows both spin and charge fluctuations. The interaction in the Kondo model is the exchange interaction, while the Anderson model has both Coulomb and hybridization interactions. The "Kondo interaction" in this paper is used rather carelessly without definition.

The authors argue that crystal electric field (CEF) effect and phonons may be candidates for the universal scale T_v . However, since the CEF splittings and phonon spectra should vary depending on materials, the argument is not convincing.

In summary, this paper has the critical weakness in the fundamental part. The referee does not recommend the paper for publication.

We would like to thank the three reviewers for their careful reading of our manuscript and their valuable comments. They were extremely helpful in identifying the weak passages of our manuscript. In the revised version of our manuscript we tried to well account for all comments and issues raised by the reviewers. We are confident that the reviewer will find the revised version much improved and clearer in the presentation of our main findings. Below we give a point-by-point reply to all comments made by the three reviewers.

Reply to Reviewer #1

We would like to thank Reviewer #1 for making us aware of some of the short comings in our previous manuscript. Namely, we realised that the we should expose more of the details of our data analysis to the reader (new Supplementary Sections 1 and 2) and need to present the data in Fig. 3 in a clearer manner. Below we answer all questions raised by the Reviewer in detail.

Referee: (1) The $\Delta n_h(T)$ data are non-saturating at high temperatures. How can we be sure about the fits to the data?

Reply: The Reviewer rightly points out a weakness in the presentation of these fits in Fig. 3 of our initial submission. The same point has also been brought up by Reviewer #3 and we revised the figure accordingly. **In general, it should be noted that our results are independent of how exactly T_v is defined and extracted from the experimental data.** The observation that the temperature scale for valence changes does not correlate with T_K , is similar for all studied Kondo lattices and also coincides with what has been reported in the literature for other Yb compounds can be read directly from Fig. 3b without any detailed analysis or fits to the data. Figure R1 shows the experimental curve for YbCo₂Si₂ and YbRh₂Si₂ where T_K differs by two orders of magnitude. The curves are literally on top of each other. Even if we would include the literature data for YbNi₃Ga₉ with a reported T_K more than 20 times higher than that of YbRh₂Si₂ and 2000 times higher than that of YbCo₂Si₂ the temperature scale on which the valence changes occur remains the same and does not reflect T_K . This statement is valid independent of any fit or further analysis.

In the original manuscript, the analytic form used for fitting the low to room temperature data is shown in Fig. 3 as dashed lines extending up to high temperatures above 1000 K, beyond the range at which experimental data is available. Clearly, the agreement at low temperature is very good, but we agree that at temperatures further above the experimental window this does not need to be the case and the plotted lines are speculative. We now show them only in the temperature for which data is available. It should be stressed that the data analysis and discussion in the paper is not affected by the details of the evolution far above room temperature. As we state in our manuscript, "*we arbitrarily identify T_v [the temperature scale for the valence changes] with the temperature at which $\Delta n_h(T) = 1/2\Delta n_h(0)$ following*

Figure R1: Temperature dependence of $\Delta n_h(T)$ in YbCo_2Si_2 , YbRh_2Si_2 and YbNi_3Ga_9 normalised to the low temperature value $\Delta n_h(0)$.

[30].” This definition does not depend on which expression is used to fit to the data, but only asks where the experimental curves in Fig. 3b of our manuscript would cross the line where the reduced valence has dropped to a value of 0.5. All of the experimental curves reach or almost reach that line in which case only little extrapolation is needed. Therefore, the uncertainty in extracting T_v from our experimental data is very small and not depending on the details of the evolution at very high temperatures.

As we mentioned, this definition of T_v is somewhat arbitrary but consistent with previous literature so that our reported values of T_v can be immediately compared with other reports in the literature. Another way of defining the temperature scale could be to consider the onset of valence changes. This onset could, for instance, be defined as the point in Fig. 3b where the constant line at $y = 0$ and a linear function approximation the region with constant slope at higher temperatures cross (see Figure R2). In this case, the found T_v values would all be around 30 K, i.e. well within the temperature range of the experimental data. The T_v obtained in this way differ from those reported in the paper only by a constant factor, and the experimental result shown in Fig. 4a and the related discussion would not change (cf. Figure R2).

Referee: (2) A related issue that probably causes some uncertainty is that while the raw RXES data is sharp at low temperatures, it presumably broadens at high temperatures. How robust is the deconvolution of the data into the 2+, 3+, and background components at high temperature? Should there be error bars on n that grow with increasing T ? How does this change fits to $\Delta n_h(T)$?

Reply: Temperature broadening effects would be of the order of tens of meV while the natural width of the features in the spectra is several eV given by the Yb 3d core-hole lifetime. Therefore, thermal broadening is not important for the spectral width here and low

Figure R2: (a) Straight line fits to the high temperature part of the experimental curves. (b) T_v values from the straight line fits using once the temperature at which y cuts zero and $y = 1/2$, respectively (semitransparent circles). The values found for $y = 1/2$ agree with those reported in the paper (opaque circles with errorbars) within the errorbars. Using the onset of temperature changes, i.e. the cut of the straight line with $y = 0$, yields for all compounds T_v values of about 30 K which is well within our experimental data range. The observation of a universal temperature scale independent of T_K does not depend on how exactly T_v is defined or extracted.

and high- T data are equally sharp. This would have been much clearer if we presented the fits to the raw data also at high temperatures which we clearly missed. In the Supplementary Information we now show for YbRh_2Si_2 (relatively large variation and deviation from Yb^{3+}) and for YbCo_2Si_2 (very small deviation from Yb^{3+}) the experimental data and fits over the entire temperature range. For the fit, we modelled the spectra with one component for Yb^{2+} and one for Yb^{3+} plus quadrupole excitations, $E2$. The shape of these two components was fixed in the fits and the same for all samples and temperatures. It was determined from the spectra itself. The only free variables was the amplitude and energy position for each of these two components. A linear function was used to model the background. This gives 6 free parameters in total of which 2 define just a linear background. The fit is extremely robust for all samples and temperatures. Animated by the comment of the referee we now added errorbars to the datapoints in Fig. 3 that correspond to the 2σ confidence interval of the $2+$ and $3+$ amplitude propagated forward through the expression used to determine the valence. The size of the errorbars does not increase with temperature and is within the symbol size for the data shown in Fig. 3a.

Referee: (3) Just looking at Fig. 3b by eye, there is some variation in the temperature dependences of different materials. To what extent does the logarithmic x-axis submerge actual differences between the various materials?

Reply: The logarithmic temperature plot is the natural choice when discussing Kondo physics with its slowly varying, logarithmic T dependences as well as when comparing compounds

Figure R3: Fig. 3b of the manuscript replotted on a linear scale.

with characteristic scales varying over several orders of magnitude. Therefore, in any textbook or review dealing with the temperature scaling of the Kondo model the log scale is used in plots [see for instance Ref. [1] and [2] of Reviewer #2 or Ref. [30] of our manuscript]. In Figure R3 we replot Fig. 3b on a linear scale. The conclusions remain the same independent of how the data is plotted, namely that **while T_K varies over many orders of magnitude the temperature scale for valence changes by less than a factor of two.**

Referee: If these points can be solidified, then there is still a question as to whether the proposed mechanisms for the anomalous temperature dependence are the right ones to consider. As stated by the authors, there is little work to help address this question. This makes it hard to assess the robustness of this discussion and makes it appear speculative.

Reply: It is obvious from the comments of all three referees that a clear, unambiguous explanation of our experimental results is desirable. However, as for instance our discussion with Reviewer #2 shows, this is a hard theoretical problem that is not easily addressed with the current machinery. We understand our paper as a **clear experimental demonstration** that the temperature scale for valence changes in Kondo lattices does not correlate with the Kondo energy scale. Surely this surprising and unintuitive result will trigger further, more detailed theoretical work on the temperature dependence of the f -level occupation in Kondo lattices in order to reconcile our experimental findings and data reported in the literature with theory. We therefore, despite the lack of clear explanation, consider this manuscript as very important in the field. The included discussion of possible mechanisms does not claim to give a definite answer but is merely meant to stimulate further theoretical work. The scope of our paper as a report of a surprising but very clear experimental result is already outlined in the first words of the title, **“Observation of ...”**.

Finally, we would like to acknowledge Reviewer #1 for the detailed investigation of our manuscript. The comments helped us understanding where we missed to display some

important technical details related to our studies. Those are now shown in the Supplementary Information file or in the revised version of the manuscript.

Reply to Reviewer #2

We are very pleased that our study got positive assessment by Reviewer #2 who considers our work “of interest to the readership of *Nature Communications*.”. The Reviewer has provided us with very helpful comments mostly related to the discussed physics. We particularly realized that we needed to better explain that the subject of our study are compounds which are well within Kondo regime, and how that is different to the mixed valent regime. This has clearly not been done well enough in the previous version and misled the Reviewer, for which we would like to apologize. Particularly, we added Supplementary Section 3, in which we summarize the relevant parameters of the Anderson impurity model and some predictions for the Kondo regime to which we refer in the manuscript. Below we carefully reply to each of the questions put forward by Reviewer #2.

Referee: 1. The authors claim at the bottom of page 4 that “the line shape of Yb 3+ and Yb 2+ lines is not symmetric”. How did the authors fit the data in Fig. 2? Did they use a skewed Gaussian and fitted the amount of skewing? Can they be sure that this procedure does not affect the numerically sensitive integral under the RXES curve in Fig. 1 and Fig. 2? In other words, what is the analytical profile of the curve used to fit Yb 3+ and Yb 2+ components in Fig. 2a and how accurately has it been determined?

Reply: The line shapes for the Yb²⁺ and Yb³⁺ component has been determined from the difference in the experimental spectra of YbRh₂Si₂ at 3K and 300K. The procedure is now comprehensively described in the Supplementary Information. The shapes used in the fit are described by the symmetric and asymmetric Pseudo-Voigt lines for the 2+ and 3+ component respectively. The line shapes of the 2+ and 3+ component were fixed, i.e the same for all temperatures and compounds, and the only free parameters were their amplitudes and position. The background was modeled by a linear function. The least-squares fitting of all spectra using this model was numerically very robust and was done automatically by a script with no manual adjustments. We now show as examples the results of the fits for YbRh₂Si₂ (large deviation from 3+ and large variation) and Yb(Co_{0.68}Rh_{0.32})₂Si₂ (small deviation and variation) at all temperatures in the Supplementary Information.

Referee: 2. Somewhat related: can the authors place the errorbars on the datapoints of the valence displayed in Fig. 3 and 4 as a result of numerical uncertainties in fitting the RXES spectra by (asymmetric) Gaussian-like curves? I was under the impression that providing the errorbars was a necessary prerequisite for publishing experimental data in the Nature family journals, is it not? I would imagine that the errorbars should be the larger, the smaller the

value of $\Delta n_h(T)$ is, since the extraction of the valence relies on extracting a tiny shoulder in the Yb 3+ curve, as is most pronounced in the case of YbCo_2Si_2 in Fig. 1

Reply: We added 2σ errorbars to the data points in Fig. 3 which we calculated from covariance matrix of the fits of the 2+ and 3+ amplitudes in the experimental spectra. The absolute error for $\Delta n_h(T)$ is comparable for all compounds but as the referee states the relative error increases with smaller values of $\Delta n_h(T)$ as can be seen in the new Fig. 3b, after rescaling the curves to their low T values. However, even for the smallest values of $\Delta n_h(T)$ they stay within $\pm 5\%$. In Fig. 3a the errorbars due to the fitting procedure remain within the symbol size. In Figure R4 we replot Figure 3a without markers in order to show the errorbars.

Figure R4: Errorbars for the data points in Figure 3a of the manuscript.

Referee: 3. On page 5, the authors make a vague statement that determining the ratio of Yb 3+ and Yb 2+ integrated intensities hinges, crucially, on the fact that, I quote "the Yb 3+ emission line is reduced to almost a tenth of its maximum intensity when the incident photon energy is set to Yb 2+ resonance of the L3 absorption edge." Can the authors please rewrite this sentence in a way that is legible by a non-expert? Do they mean that the integrated value under the Yb 3+ line should really be multiplied by a factor of about ten? Is that what has been done to plot the data in Figs. 1 and 2? Why are the authors vague about the reduction factor? "almost a tenth" sounds uncertain. I could argue that one ninth is also "almost a tenth," so this would imply that there is a systematic 10% error in all the data? If the extraction of the valence hinges on this factor, it is important that the authors clarify it.

Reply: The precise number for the relative reduction of the Yb^{3+} line in the peak of the Yb^{2+} resonance is 13% and discussed in our paper [Phys. Rev. B **84**, 245114 (2011)] which we quote at the end of the sentence. We rephrased this sentence in the manuscript, now giving the more precise number. A significant error on this number would slightly rescale all the curves in Fig. 3a and thus slightly move the datapoints in Fig. 4 on the y axis, but beyond that it would not affect the findings of our paper.

Referee: 4. Can the authors please formally define the quantity Γ first introduced at the bottom of page 8? I assume they meant $\Gamma = \pi |V|^2 N(E_F)$, defined in terms of the hybridization V in the Anderson model, and $N(E_F)$ is the conduction electron DOS, is this correct? I would like to point out that in many other literature sources, this quantity is denoted by Δ and is referred to as the "hybridization strength." The authors should be wary of this and not assume that the reader will automatically understand the meaning of .

Reply: The notation Γ for the hybridization strength is quite common in the context of Kondo systems containing lanthanide and actinide ions. In fact, the review by P. Schlottmann cited by the referee uses exactly this notation (see Eq. (2.2)). In f -electron systems, the letter Δ is often used for Crystalline Electric Field excitation energies. In order to avoid confusion, we added a definition of the parameters of the Anderson Model and of Γ in the new Supplementary Information, Part 3.

Referee: 5. I have a conceptual problem understanding the statement that " T_K itself depends exponentially on Γ " on page 8. You see, this is only true in the extreme limit of vanishing $U \rightarrow 0$ in the Anderson model, when Γ is at its bare value. Otherwise, the "hybridization width" becomes a running coupling constant, dependent on the energy scale, and some authors denote it by $\tilde{\Gamma}$ (or $\tilde{\Delta}$ in the notation of Alex Hewson), to distinguish from the bare value of Γ . In particular, in the "Kondo limit" $U \rightarrow \infty$, it is well known that the hybridization width is **not** given by the bare width $\Gamma = \pi |V|^2 N(0)$ (this would be huge, of the order of ≈ 50 meV!); instead, $\tilde{\Gamma} \rightarrow k_B T_K$ (I refer the authors to Ch. 5 in Hewsons book [1], where this is discussed in detail). So, the experimentally observed "hybridization width" is heavily renormalized and is actually *linearly* (not logarithmically!) dependent on the Kondo temperature T_K in this limit. I realize that the real system is not in the Kondo limit $U \rightarrow \infty$, but in some intermediate U regime; nevertheless, I would argue that the real Yb compounds are closer to the large- U limit than they are to the non-interacting limit. Therefore, it is not clear why the authors make a claim that the hybridization width is exponentially dependent on T_K . I urge the authors to re-evaluate their statement and to denote unambiguously which "hybridization width" they meant (and why).

Reply: We apologize that we did not define Γ precisely. Here we meant the bare hybridization strength. We fully agree with the referee that the Yb-compounds under consideration are more in the large- U -limit. In this limit, however, the Kondo temperatures for the symmetric

as well as the asymmetric Anderson model are respectively given by the expressions

$$k_B T_L = U \sqrt{\frac{\Gamma}{2U}} \exp \left\{ -\frac{\pi U}{8\Gamma} + \frac{\pi \Gamma}{2U} \right\} \quad \text{and} \quad k_B T_L = U \sqrt{\frac{\Gamma}{2U}} \exp \left\{ -\frac{\pi |\epsilon_f| |\epsilon_f + U|}{2U\Gamma} \right\}$$

(see Hewson book, Eqs. 6.109 and 6.115), which both display the exponential dependence of the characteristic temperature on the **bare** hybridization strength reflected in the width of the valence transition peak $f^1 \rightarrow f^0$ for Ce systems and $f^{13} \rightarrow f^{14}$ for Yb systems which are centered at ϵ_f . However, while the fundamental model parameters of the Anderson model, the energy ϵ_f and the bare hybridization strength Γ , can be estimated to some extent from photoemission experiments in the case of Ce compounds they would require information on the BIS side in the case of Yb compounds.

The renormalized quantities such as the hybridization width $\tilde{\Gamma}$, the renormalized f -level position $\tilde{\epsilon}_f$, and the renormalized interaction \tilde{U} mentioned by the referee can be used to characterize the Fermi liquid state which develops in the extreme low-temperature regime $T \rightarrow 0$ where the lanthanide impurity behaves like a non-magnetic yet strongly repulsive impurity. The scope of the present paper, however, is the cross-over from the relatively weakly correlated regime at high T to the strongly correlated Kondo regime at low T which cannot be discussed in terms of the renormalized Fermi liquid quantities. In the Kondo regime which is relevant for our study the statement, “ T_K itself depends exponentially on (the bare hybridization) Γ ”, is in accordance with standard textbooks on Kondo physics.

Referee: 6. Related to the above issue, the statement that “in a simple approximation, $\Delta n_h(0)$ is proportional to T_K/Γ ” is questionable. Can the authors please provide proof of this statement in either the Appendix or Supplementary Materials?

Reply: The quoted relation between the valence change due to spin fluctuations and the Kondo temperature in the low-temperature limit can be seen rather easily using e.g. the simple approximation to the ground-state wave function suggested by Gunnarsson-Schönhammer [Phys. Rev. B **28**, 4315 (1983)] for the Anderson impurity with orbital degeneracy N_f , or by the slave-boson mean-field approach for the Kondo lattice (see e.g. Hewson book Eq. 7.104)

$$n_f(T=0) = \frac{N_f \Gamma / \pi T_K}{1 + N_f \Gamma / \pi T_K} \rightarrow 1 - \frac{\pi T_K}{N_f \Gamma}$$

which implies

$$1 - n_f(T=0) = \Delta n_f(0) \simeq \frac{\pi T_K}{N_f \Gamma}$$

as stated in the paper. We have added a respective paragraph to the Supplementary Information.

Referee: 7. I take an issue with the authors’ statement on page 9 that the Kondo model “predicts the T scale of the recovery of the 3+ state to be proportional to T_K , an intuitively

trivial relation.” I disagree, based on the following argument. Let me quote the exact result, based on the Friedel sum rule, that the occupancy of the f-level per spin channel is given by (see Eq. (5.47) in Ref. [1]):

$$\langle n_{f,\sigma} \rangle = \frac{1}{2} - \frac{1}{\pi} \arctan \left(\frac{\tilde{\epsilon}_f}{\tilde{\Gamma}} \right) \quad (1)$$

where I have used the same notation as is standard in the Yamada-Yosida approach to the Anderson model (summarized in Ch. 5.3 of Hewsons book [1]), modulo denoting the hybridization by $\tilde{\Gamma}$ instead of Δ :

$$\tilde{\Gamma} = z\Gamma \quad (2)$$

$$\tilde{\epsilon} = z (\epsilon_f - E_F + \Sigma_\sigma^R(E_F)), \quad (3)$$

$$z = (1 - \partial \Sigma_\sigma^R(E_F) / \partial \omega)^{-1} \quad (4)$$

It is difficult in general to obtain analytic expressions for the above quantities, however one approach that appears fruitful to me, is to use the poor-mans RG scaling for the Anderson model, where some quasi-exact results can be obtained. In particular, I would like to quote the expression for $\tilde{\epsilon}_f$ obtained specifically *in the mixed-valent regime*, from Eq. (3.73) in Hewsons book:

$$\tilde{\epsilon}_f = \epsilon_f + \frac{\Gamma}{\pi} \ln \left(\frac{\pi D}{2 \max(\Gamma, k_B T)} \right). \quad (5)$$

The above equation captures the effect of the thermally excited charge fluctuations on the impurity site (i.e. mixed valence), which is what we are after. Substituting this into Eq. (1), one observes, qualitatively, that the change in the valence should occur at the temperature scale T_v when $k_B T_v \approx \Gamma$. Note that T_v is proportional to the bare hybridization scale $\Gamma = \pi |V|^2 N(E_F)$ of the Anderson model, and not the renormalized hybridization width $\tilde{\Gamma}$, and *not* the Kondo scale!

The key point is that the bare energy scale Γ is roughly constant across multiple Yb compounds and does not depend sensitively on the position of the f-level or the Kondo temperature. One concludes that the change in the valence from 3+ for Yb should occur at the temperature scale T_v that is roughly constant, of the order of $T_v \sim 500$ K (under the realistic estimate $\Gamma \sim 50$ meV). And this is exactly what the authors have found!

Of course I realize that the RG scaling argument is approximate and the authors may question the above derivation of Eq. (5). In its defense, I would like to show the data for n_f obtained by Pedro Schlottmann based on some exact results obtainable by Bethe ansatz, see Figure below. The different curves correspond to different bare positions ϵ_f of the f-level; in particular curves (c) and (d) are representative of the mixed valent regime. The key point is that regardless of exactly which curve one focuses on, the change in valence occurs at the temperature scale $T_v/\Gamma \sim 1$, proving my earlier point that T_v is of the order of the bare hybridization scale Γ and *not* the Kondo scale.

Figure R5: Dependence of the f -level occupation number on the temperature in $N = 6$ Anderson model. Reproduced from Fig. 5.1 in Ref. [2]. Curves (c) $\epsilon_f = 0$ and (d) $\epsilon_f = -3\Gamma$ are representative of the mixed valent regime for Yb systems.

Reply: First, we would like to emphasize that the Yamada-Yosida theory is a renormalized Fermi liquid theory derived for the extreme low-temperature limit $T \rightarrow 0$. The generalized Friedel sum rule (Hewson book, Eq. 5.47) as well as the definitions of the renormalized quantities refer to the case $T = 0$ which is explicitly stated in Hewson's book in Chapter 5.2 in the paragraph preceding the Friedel sum rule (*"We use certain exact results for the self-energy at $T = 0$ in this and subsequent sections"*).

The Friedel sum rule - even with a temperature-dependent renormalized f -level energy - cannot account for the variation with temperature of the f -valence for the following reason: Being based on a Fermi liquid picture it uses a simple resonant-level form for the f -spectral function instead of the complicated multi-peak spectral function. For this reason, the Friedel sum rule approach cannot account for the transfer of spectral weight from the high-energy regime to the low-energy regime - the formation of the Kondo resonance - which is characteristic for the formation of the Kondo state.

Second, as mentioned above the hybridization strength cannot be easily determined experimentally. A rough order-of-magnitude estimate, however, can be obtained from the width of the $f^{13} \rightarrow f^{12}$ -transition following Gunnarsson-Schönhammer. Even if we account for the degeneracy of the f^{12} -configuration, the hybridization strength estimated from the width of the $f^{13} \rightarrow f^{12}$ -transition definitely exceeds 50 meV.

Finally, the variation with temperature displayed of the f -valence from Figure 5.1 of Schlottmann's review mainly results from the contribution of the charge fluctuations. On the page preceding Figure 5.1 Schlottmann explicitly states, *"In the Kondo regime, there is a strong variation only at low T on the scale of the Kondo temperature. This is a consequence of the thermal population and smearing of the Kondo resonance."* This remark refers to curve (e) where

the variation with T at low T is not resolved. **The hybridization strength sets the scale for the variation with temperature of the valence empty-level in the mixed-valent regime but not in the Kondo regime.** In the Kondo regime, as stated by Schlottman and as explicitly shown e.g. by Bickers *et al.* (Fig. 14 of our reference [30]), the scale for the variation with temperature of the f occupation is set by the Kondo temperature.

We realised that our paper was not clear, in the sense that our claim of a universal temperature scale only refers to the Kondo regime, but not to the mixed valent regime. In $\text{Yb}(\text{Rh}_{1-x}\text{Co}_x)_2\text{Si}_2$ the value of $\Delta n_h(T = 0)$ monotonically decreases from 5×10^{-2} to 5×10^{-3} when going from $x = 0$ to $x = 1$ suggesting that $\epsilon_f > 5\Gamma \dots 15\Gamma \gg \Gamma$ and that we are well within the Kondo regime. We added a paragraph to the manuscript where we explicitly state that our observation and claim of a universal T scale is limited to the Kondo regime and discuss the difference between the Kondo and the mixed-valent regime.

Furthermore we added a supplementary information where we show some data on Yb-systems well in the mixed valent regime, e.g. YbAl_2 . In this case experimental data do not show any apparent change in the valence between 3 K and 300 K, implying that the characteristic T scale for changes of the valence has shifted to far above 300 K. Thus well in the mixed valent regime, the characteristic T scale for the T induced increase in the valence is far above the value we observe in the Kondo regime, and is very likely changing from system to system. We would agree with the referee that in this regime this T scale increases linearly with the bare hybridization strength Γ , and not exponentially as T_K does. Such an increase of T_v might already affect some of the systems which we included in Fig. 3b and 4 which are at the border between the Kondo and the mixed valent regime.

Referee: To summarize, while the experimental findings presented in the manuscript are certainly interesting, I challenge the authors alleged interpretation of the valence scale T_v as being due to crystal-field effects, and I instead invite the authors to consider a much more mundane explanation, namely that the scale T_v is independent of the Kondo temperature for a simple reason that it is related to the bare hybridization scale $\Gamma \gg k_B T_K$, which is a lot less sensitive to the microscopic details of the composition, lattice constant etc.

Reply: We would like to thank the reviewer for his work and stimulating thoughts on our results. We apologize that we did not clearly state that our study and claim refers to the Kondo regime, not to the mixed valent regime, and thus mislead the referee to focus on the mixed valent regime. According to standard textbooks and review papers on Kondo physics, in the Kondo regime the T_v scale on which the occupation n_h of the f state increases with T is set by the Kondo temperature, not by the bare hybridization strength as in the mixed valent regime. Therefore our observation of nearly identical T_v values in a series of compounds where T_K decreases by almost four orders of magnitude is an unexpected result, in contradiction with prediction of standard models. Clearly, a satisfying theoretical explanation is not easy to obtain. It is the purpose of this paper to report our surprising experimental findings for the temperature dependence of the valence in Yb Kondo lattices and to show the clear need

for further theoretical work in order to understand them better. We are convinced that our results will get a lot of attention and trigger further theoretical work.

Reply to Reviewer #4

We would like to thank the Reviewer for the valuable input. We realized that the Reviewer made a number of technical comments that were probably mostly due to a misleading representation of the data. Animated by the Reviewer's comments we changed Fig. 3 in order to avoid the impression that we rely on certain speculative assumptions in our data analysis. We also made sure that we now clearly define and accurately use technical terms. We hope that the Reviewer will find the input given with his/her comments well reflected in the revised version of the manuscript and the detailed reply below.

Referee: The "universal temperature scale" is obtained by the procedure explained in Fig. 3, where the data at temperatures between 3K and 300K are fitted by a certain smooth function. The authors deduce the temperature scale T_v around 300-400 K even though no data are available above 300 K. The most serious drawback of the paper is the procedure to derive T_v . Namely

1. The authors assume trivalent Yb in the high temperature limit, which has no justification both experimentally and theoretically. Experimentally, only some increase of Yb valency with increasing temperature has been found.
2. Extrapolation to temperatures higher than 300 K relies on a function with no physical basis. Hence the value of T_v obtained around 300-400 K is not trusted, and the major conclusion does not have a scientific basis. Although the authors cite the paper by Goldhaber et al (ref.31) that uses a similar function, the quantity fitted in ref.31 is the conductance with empirical validity for interpolating the temperature dependence.

Reply: A similar question, addressing the absence of data in the high temperature range has also been raised by Reviewer #1. We would like to repeat that the details of how the temperature curves evolve far above room temperature are not really affecting our discussion. Most importantly, independent of any expression used to fit the data our experimental observation is very clear and unambiguous: the temperature dependence and temperature scale of the valence changes in the studied Kondo lattices is very similar and not at all reflecting the Kondo scale T_K . This main finding is easily read from Fig. 3b even without any fit to data.

We agree that showing the analytical expression used to fit the data from low to room temperature extrapolated to beyond 1000 K is misleading in the sense that it suggests we assume to know and use the evolution farther above room temperature to come to our conclusions. This is probably why the Reviewer concentrates on this issues and considers it

to be, “[t]he most serious drawback of the paper”. However, we clearly want to state that contrary to what the referee suggests we do not assume anything about the high temperature range but only use the range up to room temperature in our analysis.

In particular, we state in the paper that “we arbitrarily identify T_v [the temperature scale for the valence changes] with the temperature at which $\Delta n_h(T) = 1/2\Delta n_h(0)$ following [30].” This does not assume that $\Delta n_h(T \rightarrow \infty) = 0$, but is just one way of defining a characteristic temperature scale for all experimental curves. It merely assigns the temperature at which the deviation from trivalent is half that at low temperature. For the data in Fig. 3b that means extrapolating to the dashed line at 0.5 which is reached or almost reached by all of the experimental curves. Hence no extrapolation to high temperatures or assumption about the high temperature valence is necessary or used in our analysis.

Secondly, we agree that the function used to fit the data has no strong physical motivation. The reason we used it is twofold: It has a minimal number of free parameters (two) and gives an excellent fit to the data. This is not entirely surprising because being able to describe the resistivity in Kondo lattices, which somewhat scales with the number of magnetic scattering centers, suggests that it should contain information about the temperature evolution of the occupation of the $4f$ level. The main argument for using this function was that it only depends on the two parameters of interest, $\Delta n_h(0)$ and T_v . Note that for instance an arctan function would involve already three free parameters and does not describe the experimental data as well. As laid out in the discussion with Reviewer #1 we could also use another function to determine T_v . For instance, a straight line (also two free parameters) fitted to the region of linear slope will obtain very similar values for T_v . In this case, we would still need to fit the low temperature value $\Delta n_h(0)$ with a constant function, ending up again with three free parameters in total. So, the main argument for the function in the paper is its simplicity while reproducing the experimental curves excellently.

The important point is that our result is independent of the specific definition of T_v . We could, for instance, define T_v as the temperature where the valence changes set it (see discussion of point 1 of Reviewer #1). Then we would obtain a T_v around 30 K in all cases, i.e. well within the experimentally assessed temperature range. The conclusion of our paper that the temperature scale for the valence changes does not reflect T_K and that it is very similar for Kondo lattices spanning several orders of magnitude in T_K would not be affected at all. This should become immediately obvious from Fig. 3b. Below we show again Figure R1 with the experimental curves for YbCo_2Si_2 with $T_K \ll 1$ K, YbRh_2Si_2 with a T_K about 100 times higher, and for YbNi_3Ga_9 where the reported T_K is 570K, more than 20 times higher than that in YbRh_2Si_2 . No analysis whatsoever is needed to see that T_K is not at all reflected in the experimentally observed T dependence. (It correlates, however, well with the low temperature value of $\Delta n_h(T)$.)

In summary, we pointed out that: (i) the observation of a similar temperature scale for the valence changes independent of T_K can be immediately read from the experimental data. (ii) We use a consistent definition of T_v in order to assign a number to the experimentally

Figure R1: Temperature dependence of $\Delta n_h(T)$ in YbCo_2Si_2 , YbRh_2Si_2 and YbNi_3Ga_9 normalised to the low temperature value $\Delta n_h(0)$.

observed curves. This definition does not assume anything about the temperature behaviour beyond the experimentally accessible range. (iii) We fitted an analytical expression with the smallest possible number of free parameters to the data, obtained excellent agreement in the entire temperature range and used it to extract T_v which requires no or very little extrapolation beyond the experimental data range. (iv) Other analytical expressions to fit the data will yield the same results as show in Fig. 4a because of the strong similarity between the curves in Fig. 3b for the different Kondo lattices. Similarly, other definitions of T_v will obviously change the value of T_v but will not change the experimental fact that it will be very similar in all cases and definitely not reflect the variation in T_K by several orders of magnitude. We therefore cannot understand the reviewer's comment that "the value of T_v obtained around 300-400 K is not trusted, and the major conclusion does not have a scientific basis."

Referee: Other remarks: The analysis in this paper using the terminologies of "Kondo model" and the "Kondo interaction" is very confusing. In the standard usage, the Kondo model refers to the case where charge (valence) fluctuations are projected out. If the valence fluctuation is to be included explicitly, the Anderson model for example should be used that allows both spin and charge fluctuations. The interaction in the Kondo model is the exchange interaction, while the Anderson model has both Coulomb and hybridization interactions. The "Kondo interaction" in this paper is used rather carelessly without definition.

Reply: We apologize that we were sloppy with the term "Kondo interaction". We now included a definition of the Kondo interaction as a hybridization induced exchange, mention the charge fluctuation in the intermediate valent regime as well as the Anderson model. The valence in all materials considered is close to the integer value 3. From this observation we conclude that charge fluctuations play only a minor role. The variation with temperature of the valence due to spin fluctuations is colloquially ascribed to the "Kondo effect".

Referee: The authors argue that crystal electric field (CEF) effect and phonons may be candidates for the universal scale T_v . However, since the CEF splittings and phonon spectra should vary depending on materials, the argument is not convincing.

Reply: One certainly expects different Yb-based compounds to present very different CEF scheme, and some differences in the Debye temperature. However, the CEF energy scale, i.e. the energy difference between ground and highest excited CEF level do not differ so much between different Yb compounds and are typically in the range 200 K to 400 K. Thus both the absolute value and the range of the overall CEF splitting are compatible with the value and range of T_v . This is in contrast with the Kondo scale, which differs by almost 4 orders of magnitude in the investigated compounds. As already stated in the first version of the paper, a similar argument applies to the Debye temperature. Therefore the expected difference in the CEF as well as the expected difference in the Debye temperature should not lead to a stronger scatter in T_v than that which we observed. Therefore these expected differences cannot be taken as arguments against CEF or lattice vibration playing a role. We added two sentences where we mention and discuss the possible influence of different size of the CEF splitting.

Referee: In summary, this paper has the critical weakness in the fundamental part. The referee does not recommend the paper for publication.

Reply: The criticism raised by the referee has been of technical nature. In point 1, the referee raised concerns regarding the validity of our approach for extracting T_v from the spectra. We showed in our reply that our result (i) is independent of how T_v is extracted, and (ii) can immediately be read from the data shown in Fig.3 of our manuscript even without assigning a value T_v to the experimental curves. Point 2 criticises that we were not clear enough about what we mean when we use the terms “Kondo model” and “Kondo interaction.” The definition of the term “Kondo interaction” and its distinction from the valence fluctuating regime is now explicitly given in our manuscript. In point 3, our discussion that excited CEF levels or lattice vibrations may be important for the observed universality is criticised as speculative and not compatible with the spread of these properties in between the different compounds. We clearly marked these two effects as propositions with a reasonable physical foundation in the paper. As laid out in our reply and now also in the respective section of the supplementary information, the differences in Δ_{CEF} and T_D should not lead to a larger spread in T_v than what we observe. Hence, all three technical issues brought up by the referee can be readily addressed.

How these technical questions justify seeing a critical weakness in the fundamental part of our work in the first place is not clear to us. The mission of this manuscript is to report an unexpected **experimental observation** in a highly relevant field. This mission is clearly outlined already in the title. At no point did we claim to also present a rigorous theoretical explanation for our results which will clearly require future work. What we show is that the currently employed simple approaches to the Kondo model must be incomplete as they fail

to explain our findings. We also show how our results can be immediately useful to future experimental studies in that they present a way of determining the Kondo scale with very high accuracy even in the presence of AFM correlations which has been an open experimental problem up to now. Consequences for other highly relevant questions in the field of Yb Kondo lattices immediately arise from our results. As an example, we now explicitly discuss that our results, for instance, exclude the presence of a valence quantum critical point in YbRh_2Si_2 , a scenario previously put forward to explain very unusual thermodynamic, transport and magnetic properties close to zero temperature.

In summary, we present a carefully conducted, comprehensive experimental study with clear results that are highly relevant for assessing and improving our understanding of $4f$ Kondo lattices. We are therefore convinced that our manuscript is suitable for publication in Nature Communications.

Reviewers' comments:

Reviewer #1 (Remarks to the Author):

I believe that the authors have carefully responded to the points 1-3 that I raised. This resulted in revisions to the manuscript and supplementary material that clarify (1) the presentation of the data and (2) the arguments that are made based on the data. I am now satisfied with this part, and furthermore agree that the scaling (and lack of scaling) relationships presented in Fig. 4 are convincingly arrived at. Figure 4 presents an interesting and novel outcome that I believe that this will be of interest to many researchers, especially in the area of strongly correlated f-electron metals.

As the authors acknowledge, an important factor that is missing is an unambiguous explanation for the observed trends. Perhaps they are correct that this initial observation will trigger further theoretical work. As they state, this is a challenging problem. Therefore, I would recommend this article for publication, with the reservation that despite the fact that the authors speculate about possible explanations for their data, there still isn't a clear understanding of what is observed.

Reviewer #2 (Remarks to the Author):

I thank the authors for carefully responding to my criticisms and for clarifying that the bulk of the data have been obtained in the Kondo regime, where the valence deviation from $3+$ is small, and not in the mixed-valent regime, for which I have provided detailed reasoning. I also welcome the authors having extended the Supplementary Information to include the theoretical details and to specifically answer my concerns.

Having read the revised manuscript, I would like to recommend it for publication in the Nature Communications, *provided* the following (minor) change to the main text is made first: I would like the authors to explicitly include the statement similar to the one they made in response to my comments, I quote:

"We would agree with the referee that in the mixed valent regime, this characteristic T scale [of the valence change] increases linearly with the bare hybridization strength Γ , and not exponentially as T_K does. Such an increase of T_v might already affect some of the systems which we included in Figs. 3b and 4, which are at the border between the Kondo and the mixed-valent regime."

An explicit acknowledgement of the mixed-valent regime is all the more important, given that the authors chose to display the data for YbAl_3 and YbNi_3Ga_9 in Figs. 3b and 4, which are much closer to the mixed-valent scenario than they are to the Kondo regime. My main criticism was precisely that in the mixed-valent regime, one expects T_v to scale linearly with Γ (rather than T_K), and this may explain the data for at least YbAl_3 and YbNi_3Ga_9 . Since the authors explicitly mention YbNi_3Ga_9 in the main text, it is only fair that they should also include a careful statement about how T_v is affected by the real valence fluctuations, as they did in reply to my comments.

Reviewer #4 (Remarks to the Author):

I recognize significant improvement on presentation. I agree with the authors' statement that T_v , which characterizes the change of $n_f(T)$, is not always related to T_K . As admitted by the authors, however, the value of T_v for a given Yb system depends on the definition, and ranges from $O(10)$ K to $O(10^2)$ K as shown in Fig.R2. The estimate by intersection with the straight line as in Fig.S4 is more reasonable than the arbitrary use of 0.5 in the normalized variation.

On the other hand, the main claim of the paper is not justified about the universality of T_v in the

Kondo regime of Yb systems. A counter example is YbNi₃Ga₉, which is strongly mixed valent (~2.4), but has a similar T_v as shown in Fig.R1. Another example is YbAlB₄, which is also out of the Kondo regime (valence ~2.7) but has similar T_v according to Fig.S4. The authors emphasize the difference of slope in YbAlB₄, but the intersection is below 100K as in other Yb 122 systems. Furthermore, YbAl₂ has much larger T_v as remarked by the authors. Hence there is no systematics about T_v whether the system is in the Kondo regime or in the mixed valence regime.

The change of $n_f(T)$ in the lattice system has the restriction that the total number of electrons be a constant independent of T . In this sense, theoretical results for the Anderson impurity model with constant f -electron level cannot always be applied to temperature dependence in lattice systems. This point seems relevant in discussing discrepancy from results for impurity models, in addition to other complexities discussed by the authors.

In summary, one of the authors' assertions that T_v is not related to T_K in certain Yb systems seems valid. On the other hand, the main assertion of universality of T_v in the Kondo regime is not justified experimentally, although it may apply to the particular tetragonal structure of Yb systems. Since the paper is misleading in this respect, I do not recommend publication of the paper in the current form.

We would like to thank the reviewers for evaluating our revised manuscript and our reply to their comments. We were pleased to read that all three reviewers felt that we used their comments to significantly improve our manuscript. None of them raised major objections against publication of our manuscript in Nature Communications but some minor issues were raised that should be addressed. In reaction, we have further revised our manuscript accordingly and added a new section 5 to the Supplementary Information. The changes to the manuscript are highlighted in blue color. Below we give a detailed reply to the report of each reviewer.

Reply to Reviewer #1

Referee: I believe that the authors have carefully responded to the points 1-3 that I raised. This resulted in revisions to the manuscript and supplementary material that clarify (1) the presentation of the data and (2) the arguments that are made based on the data. I am now satisfied with this part, and furthermore agree that the scaling (and lack of scaling) relationships presented in Fig. 4 are convincingly arrived at. Figure 4 presents an interesting and novel outcome that I believe that this will be of interest to many researchers, especially in the area of strongly correlated f-electron metals.

As the authors acknowledge, an important factor that is missing is an unambiguous explanation for the observed trends. Perhaps they are correct that this initial observation will trigger further theoretical work. As they state, this is a challenging problem. Therefore, I would recommend this article for publication, with the reservation that despite the fact that the authors speculate about possible explanations for their data, there still isn't a clear understanding of what is observed.

Reply: We would like to thank the referee for reviewing our manuscript again and were pleased to read the positive feedback on our reply and the revised manuscript. As the referee points out, despite proposing some possible explanations, we do not have a clear theoretical understanding of our experimental observations. At several places in our manuscript we therefore stressed that further experimental and theoretical work is needed in order to get a better understanding of our unexpected results. We have now also added a line to the abstract that explicitly states that, *"This observation is in remarkable contradiction to both naive expectation and precise theoretical predictions of the Kondo model, asking for further theoretical work in order to explain our findings."* and don't mention excited CEF levels and lattice vibrations in the abstract anymore. Similarly, we end our discussion now with the statement, *"Looking for the origin of this unexpected universal T scale we identify two possible candidates, excited CEF levels and lattice vibrations. However the effect of both properties on $n_h(T)$ has not been studied yet and is thus presently not clear. Therefore our results urge for more experimental and theoretical studies on how excited CEF levels and lattice vibrations affect the T dependence of the valence. Without respective information*

available it is not clear whether excited CEF levels or lattice vibrations can explain our observations, or whether another mechanism is active. The very similar temperature scale T_v observed in all compounds in this study gives, at least, a good idea of the energy scale of the relevant mechanism." This should make it clear to any reader that we only speculate about possible explanations for our experimental observations which call for future theoretical work. In this context we would like to remark that the question how a Kondo lattice evolves upon increasing temperature starts to attract considerable attention. Recently a few papers e.g. addressed the questions at which temperature and how the transition from a low T coherent Fermi liquid state with composite heavy fermions to an incoherent high temperature state with decoupled $4f$ and conduction electrons takes place, making obvious that this question is quite complex and far from being understood. Here knowledge on how n_f evolves as a function of T in different systems with largely different T_K is an important input.

Reply to Reviewer #2

Referee: I thank the authors for carefully responding to my criticisms and for clarifying that the bulk of the data have been obtained in the Kondo regime, where the valence deviation from $3+$ is small, and not in the mixed-valent regime, for which I have provided detailed reasoning. I also welcome the authors having extended the Supplementary Information to include the theoretical details and to specifically answer my concerns.

Having read the revised manuscript, I would like to recommend it for publication in the Nature Communications, *provided* the following (minor) change to the main text is made first: I would like the authors to explicitly include the statement similar to the one they made in response to my comments, I quote:

"We would agree with the referee that in the mixed valent regime, this characteristic T scale [of the valence change] increases linearly with the bare hybridization strength Γ , and not exponentially as T_K does. Such an increase of T_v might already affect some of the systems which we included in Figs. 3b and 4, which are at the border between the Kondo and the mixed-valent regime."

An explicit acknowledgement of the mixed-valent regime is all the more important, given that the authors chose to display the data for YbAl3 and YbNi3Ga9 in Figs. 3b and 4, which are much closer to the mixed-valent scenario than they are to the Kondo regime. My main criticism was precisely that in the mixed-valent regime, one expects T_v to scale linearly with Γ (rather than T_K), and this may explain the data for at least YbAl3 and YbNi3Ga9. Since the authors explicitly mention YbNi3Ga9 in the main text, it is only fair that they should also include a careful statement about how T_v is affected by the real valence fluctuations, as they did in reply to my comments.

Reply: We would like to thank the referee for the helpful discussion and for carefully examining our reply to the previous comments. Animated by the discussion with the referee

during the first review round we had already included a respective statement about the expected scaling between T_v and Γ , in the Supplementary Information, section 3, of the revised manuscript. But we agree that this should at least briefly also be mentioned in the manuscript. Following the referee's advise we now also added a line to the main manuscript discussing the different scaling of T_v with Γ in the Kondo and in the strongly mixed-valent regime.

Reply to Reviewer #4

Referee: I recognize significant improvement on presentation. I agree with the authors' statement that T_v , which characterizes the change of $n_f(T)$, is not always related to T_K . As admitted by the authors, however, the value of T_v for a given Yb system depends on the definition, and ranges from $O(10)$ K to $O(10^2)$ K as shown in Fig. R2. The estimate by intersection with the straight line as in Fig.S4 is more reasonable than the arbitrary use of 0.5 in the normalized variation.

Reply: We agree with Reviewer #4 that the definition of T_v we used in our manuscript is not the only possible one. One might also use the onset of temperature dependence as we showed in our previous reply, Fig. R2. Then one obtains an absolute value that is of the order of 30 K for all measured Kondo systems, rather than the 300 K that we get with the definition used in the paper. However, this difference of one order of magnitude is irrelevant for our main claim which, in its essence, is not about a specific absolute value of T_v but about the evolution of T_v upon changing T_K . And here our experimental results unambiguously demonstrate that changes in T_K are not reflected in changes in T_v . Instead T_v is very similar among the investigated systems even though T_K changes by orders of magnitude. As we have already shown in our previous reply, this claim is not only well supported using our definition of T_v , it is also well supported using the definition proposed by Reviewer #4. Hence, which definition of T_v is used is not important within the scope of our paper. The reason why we chose $\Delta n_h(T_v) = 0.5$ is because it is the definition used in the literature both in experimental as well as in theoretical studies. We can refer the reviewer, for instance, to Fig. 14 of Bickers et al., Phys. Rev. B 36, 2036 (1987) and to Refs. 13-16 and 19 of our manuscript. Changing the definition of T_v would not change the findings of our paper, but would complicate the comparison with previous studies and is potentially confusing when referring to theoretical work. Furthermore, in the general case that an interaction leads to a temperature driven change in a given property, the temperature at which one starts to observe this change is usually far below the characteristic energy of the relevant interaction. This characteristic energy is usually much better reflected by the temperature at which a significant part, e.g. 50%, of the change in the properties has occurred. Therefore with the definition of Reviewer #4, the absolute values of T_v would very likely be misleading concerning the energy scale of the relevant mechanism. Hence, there are good reasons for using $\Delta n_h(T_v)/\Delta n_h(0) = 0.5$ for defining T_v . We admit that potentially the line, "Here we

arbitrarily identify T_v with the temperature at which $\Delta n_h(T) = 1/2\Delta n_h(0)$ following [30]" in our manuscript may have suggested that we had no good reasons to chose this definition. We have therefore removed the word "arbitrarily" from the sentence. We also added section 5 to the Supplementary Information where we show Fig. R2 of our previous reply and discuss that other definitions of T_v , including the one proposed by the reviewer, will give different absolute numbers for T_v but that will not affect our findings. The values for T_v obtained with the definition proposed by the referee can be directly read from panel (b) of this figure.

Referee: On the other hand, the main claim of the paper is not justified about the universality of T_v in the Kondo regime of Yb systems. A counter example is YbNi₃Ga₉, which is strongly mixed valent (2.4), but has a similar T_v as shown in Fig.R1. Another example is YbAlB₄, which is also out of the Kondo regime (valence 2.7) but has similar T_v according to Fig. S4. The authors emphasize the difference of slope in YbAlB₄, but the intersection is below 100K as in other Yb 122 systems. Furthermore, YbAl₂ has much larger T_v as remarked by the authors. Hence there is no systematics about T_v whether the system is in the Kondo regime or in the mixed valence regime.

Reply: Reviewer #4 disputes the universality of T_v , but the two first counter example put forth by the reviewer (YbNi₃Ga₉ and YbAlB₄) show the same or a similar T_v as the systems in the Kondo regime, as Reviewer #4 correctly states. Therefore these two systems are in no way counter examples to our claimed universal T_v , instead they support this universal behavior, but raise the question why this universal behavior extends so far into the mixed valent regime. The third counter example given by Reviewer #4, YbAl₂, is very far in the mixed valent regime. As explicitly stated in our manuscript and in length also in our previous reply, we do not claim that the universal scale is preserved in the strongly mixed-valent regime. Instead we explicitly state in the paper that, "[...] *within the strong intermediate valent regime both the magnitude [...] as well as the T scale on which it occurs do change with the strength of charge fluctuations*", and explicitly show and discuss YbAl₂ as an examples in the mixed valent regime where the behavior is different. Therefore also YbAl₂ cannot be taken as a counter example to our claim of a universal behavior in the Kondo regime. Instead YbAl₂, as the two other systems mentioned by Reviewer #4 in this comment, raise the question how the universal T dependence that we observe in the Kondo regime evolves when one moves towards the strongly mixed valent regime. Unfortunately in this regime only very few precise RXES data have been published, and therefore a systematic analysis of this regime is not yet possible. In our opinion, in the absence of sufficient experimental data, the approach proposed by Reviewer #2, namely to discuss this problem using predictions of theoretical models, is the most appropriate.

On the other hand, from the data we show, it is fully justified to claim that within the Kondo regime the T scale for valance change is very similar in all studied systems and independent of the Kondo scale. Both Reviewer #1 and Reviewer #2 agree with this observation. In reaction to the concerns of Reviewer #4 we made sure that through-out the manuscript we only speak of "*a universal temperature scale among the investigated Kondo lattice*". We

also changed the title accordingly.

Referee: The change of $n_f(T)$ in the lattice system has the restriction that the total number of electrons be a constant independent of T . In this sense, theoretical results for the Anderson impurity model with constant f -electron level cannot always be applied to temperature dependence in lattice systems. This point seems relevant in discussing discrepancy from results for impurity models, in addition to other complexities discussed by the authors.

Reply: We thank the reviewer for raising this pertinent point. Indeed in some properties there is a huge difference between the single ion Kondo case and the dense Kondo lattice in the predicted and observed behavior. For instance, the T dependence of the resistivity at low temperatures in the Kondo lattice is completely different from that of a single Kondo ion. However these differences are based on the formation of a coherent state at low T in the lattice case, and they are therefore expected and also observed to vanish towards high temperatures. Thus it is unlikely that the difference between the single ion and the lattice plays a dominant role in the evolution of $n_f(T)$ at high T . Furthermore the temperature T_{coh} at which coherence sets in is related to the Kondo scale. In the systems we present in this study T_{coh} differs by as much as T_K . Therefore it is unlikely that the formation of a coherent state is at the origin of the nearly identical T_v in systems with such different T_{coh} . Hence the problem raised by Reviewer #4 is certainly a very pertinent one from a theoretical point of view, but unlikely to play a significant role in explaining our observation. Unfortunately, already the calculation of $n_f(T)$ including the CEF is a very hard theoretical problem. The equivalent calculation for a lattice seems to be out of present theoretical possibilities.

In order to account for this comment of Reviewer #4, we introduced Supplementary Note 5, *"We note that the theoretical calculations we refer to for the evolution of $n_f(T)$ with T_K deal with the single ion Kondo model, not the dense Kondo lattice. The latter has the additional restriction that the total number of electrons has to be constant, and therefore the position of the f level has to be shifted with T . Unfortunately a calculation of $n_f(T)$ for a Kondo lattice and including the CEF seems to be beyond present theoretical possibilities. However, differences between the single ion Kondo system and the Kondo lattice are usually associated with the onset of coherence at low T , and are therefore unlikely to significantly affect the high T regime. Furthermore the temperature T_{coh} at which coherence sets in is related to the Kondo scale and differs by as much as T_K in the systems studied here. Therefore it is unlikely that the formation of a coherent state is at the origin of the nearly identical T_v in systems with such different T_{coh} ."*

Referee: In summary, one of the authors' assertions that T_v is not related to T_K in certain Yb systems seems valid. On the other hand, the main assertion of universality of T_v in the Kondo regime is not justified experimentally, although it may apply to the particular tetragonal structure of Yb systems. Since the paper is misleading in this respect, I do not recommend publication of the paper in the current form.

Reply: Our data do not only justify the statement that T_v is not related to T_K , as Reviewer #4 concedes. In addition these data clearly show that T_v is not just erratic, but assumes a very similar value in all the Kondo lattices presented in our study, as long as they are in the Kondo regime. Demonstrating that T_v is not just erratic, but has a very similar value in all systems, independent of T_K , is a very important result. It evidences that there is a systematic, although not the one which was expected. It provides an idea about the energy scale of the relevant mechanism, and is therefore an important information to get a deeper understanding of the evolution of $n_f(T)$ with T .

The second part of the statement, “*may apply to the particular tetragonal structure of Yb systems*”, is not correct and unjustified. It is true that we start by showing measurements on a series of 122 compounds crystallizing in the tetragonal ThCr_2Si_2 structure type. But we then add a number of results on systems with completely different structures. The tetragonal structure of YbNi_4P_2 is completely unrelated to that of the 122 systems. Notably the local symmetry of Yb is orthorhombic, leading to a completely different CEF situation. YbNi_3Ga_9 , YbCu_3Al_2 and YbCu_5 are hexagonal, YbAl_3 is cubic. Therefore our observation is not limited to a specific series of compounds and one structure type. It covers a number of very different structure types, with Yb in very different local symmetries and very different chemical environment. We included all Yb-based Kondo lattices where precise T dependent RXES data are published and with a reliable determination of T_K . Having used all the available information is a sound scientific basis for proposing a universal behavior.

In order to account for this comment of Reviewer #4 we carefully checked our wording and changed the title of the manuscript to make sure that we avoid a too general statement that goes beyond our experimental data. We hope that with these revisions the referee will find our manuscript suitable for publication.